# LSCD: Lomb–Scargle Conditioned Diffusion
# for Time series Imputation

**Elizabeth Fons** [* 1] **Alejandro Sztrajman** [* 2] **Yousef El-Laham** [1] **Luciana Ferrer** [3 4] **Svitlana Vyetrenko** [1]
**Manuela Veloso** [1]

## Abstract

Time series with missing or irregularly sampled data are a persistent challenge in machine learning. Many methods operate on the frequency-domain, relying on the Fast Fourier Transform (FFT) which assumes uniform sampling, therefore requiring prior interpolation that can distort the spectra. To address this limitation, we introduce a differentiable Lomb–Scargle layer that enables a reliable computation of the power spectrum of irregularly sampled data. We integrate this layer into a novel score-based diffusion model (LSCD) for *time series imputation conditioned on the entire signal spectrum.* Experiments on synthetic and real-world benchmarks demonstrate that our method recovers missing data more accurately than purely time-domain baselines, while simultaneously producing consistent frequency estimates. Crucially, our method can be easily integrated into learning frameworks, enabling broader adoption of spectral guidance in machine learning approaches involving incomplete or irregular data.

## 1. Introduction

Time series data often exhibit missing observations or irregular sampling intervals, creating challenges for machine learning tasks such as imputation, forecasting, classification, and anomaly detection (Yang et al., 2024b). Many methods operate in the frequency domain (Alaa et al., 2021; Yang et al., 2024a; Wu et al., 2023; Crabbé et al., 2024) using the Fast Fourier Transform (FFT), which requires data to be uniformly sampled on a regular grid. To address this requirement, these methods typically handle missing values by interpolating or zero-filling the gaps prior to FFT computation. However, such pre-processing can distort the underlying data distribution, leading to spurious or attenuated frequency estimates.

By contrast, the *Lomb–Scargle* periodogram (Lomb, 1976; Scargle, 1982) can estimate power spectra without assuming uniform sampling. It does not require an imputation step to compute frequency components, making it far more reliable in the presence of irregular or missing data. Despite its advantages, Lomb–Scargle remains under-explored in the broader machine learning community.

In this paper, we introduce a *Lomb–Scargle Conditioned Diffusion* (LSCD) approach for time series imputation. Inspired by score-based diffusion methods (Song & Ermon, 2019), we develop a diffusion model that operates in the *time domain*, but is *conditioned* on a Lomb–Scargle-based spectral representation. By leveraging the full Lomb–Scargle periodogram as an additional input, our model effectively captures underlying frequency structures and imposes greater consistency between the final time-domain imputation and the original spectral content.

We evaluate our approach on both synthetic and real datasets, introducing varying degrees of missingness (up to 90%). We show that our conditioning on the Lomb–Scargle spectrum consistently improves time-series imputation metrics such as MAE and CRPS compared to purely time-domain baselines. Crucially, the method produces PSD estimates closer to the ground truth, demonstrating improved consistency of frequency recovery.

Our main contributions are:

1. We introduce a *diffusion-based time-series imputation* framework that *directly* conditions on the Lomb–Scargle spectrum, thereby eliminating the need for interpolation or zero-filling in the frequency domain. To further enhance performance with high rates of missing data, we propose a *consistency loss* that refines the alignment between the time-domain signal and its Lomb–Scargle representation.

2. We provide a differentiable implementation of Lomb–Scargle and show how it can be seamlessly integrated

---

[*]Equal contribution [1]J.P. Morgan AI Research [2]University of Cambridge [3]University of Buenos Aires [4]CONICET. Correspondence to: Elizabeth Fons <elizabeth.fons@jpmorgan.com>.

*Proceedings of the 42$^{nd}$ International Conference on Machine Learning*, Vancouver, Canada. PMLR 267, 2025. Copyright 2025 by the author(s).

into learning methods to handle missing or irregularly sampled data.

3. We provide comprehensive empirical results showing improved imputation performance and accurate frequency recovery on both synthetic and real datasets.

Beyond time series imputation, we hope our work encourages broader use of Lomb–Scargle-based techniques for irregular data within machine learning pipelines.

## 2. Related Work

Time series imputation has been extensively studied across statistical, deep learning, and spectral approaches. Early statistical methods often rely on assumptions of local continuity or cross-variable dependencies. Simple heuristics, such as mean or median imputation (Fung, 2006; Batista & Monard, 2002), are straightforward but fail to capture complex temporal dynamics. More advanced techniques, including Expectation-Maximization (EM) (Ghahramani & Jordan, 1993; Nelwamondo et al., 2007), auto-regressive models, and state-space approaches (Durbin & Koopman, 2012; Walter.O et al., 2013), incorporate structured dependencies but can struggle with high rates of missingness and irregular sampling. Gaussian Processes (GPs) (Fortuin et al., 2020) provide uncertainty-aware imputation yet face scalability hurdles and often assume smooth temporal trends.

Deep learning methods have significantly advanced time series imputation by learning long-range dependencies and complex temporal patterns. RNN-based models such as BRITS (Cao et al., 2018) and GRU-D (Che et al., 2016) incorporate explicit modeling of missingness, while Transformer-based methods like SAITS (Du et al., 2023) can better capture long-range interactions. Generative architectures, including GANs (Miao et al., 2021) and VAEs (Fortuin et al., 2020), allow sampling-based imputation under high uncertainty. More recently, diffusion models (Tashiro et al., 2021) have shown promising results by iteratively refining noisy samples, but they remain confined to time-domain representations and typically overlook the spectral properties of irregularly sampled series (Yang et al., 2024b).

Several approaches have been developed to handle irregular time-series data without requiring pre-defined fixed-length windows. MADS (Bamford et al., 2023) proposes an auto-decoding framework built upon implicit neural representations. Latent ODEs (Rubanova et al., 2019) define a neural ODE in latent space, solved using numerical methods like the Euler method, naturally accommodating irregularly sampled observations for standard time-series tasks such as imputation and generation. Extensions of latent ODEs, such as the GRU-ODE-Bayes model (Brouwer et al., 2019), handle sporadic observations with jumps, while the neural continuous-discrete state space model (NCDSSM) (Ansari

et al., 2023) introduces optimizable auxiliary variables in the dynamics for improved accuracy. Another approach considers a diffusion process over function space to handle irregularity in time-series (Biloš et al., 2023), connected with neural processes (Garnelo et al., 2018). Naturally, extensions based on neural SDEs have also been proposed, which incorporate a diffusion term in the latent dynamics (Kidger et al., 2020; El-Laham et al., 2025; Oh et al., 2024). Given that most neural ODE and SDE models require computationally expensive numerical solvers at both training and inference time, efforts have focused on developing efficient alternatives. Continuous recurrent units (CRUs) (Schirmer et al., 2022) model hidden states using linear SDEs solvable analytically with a Kalman filter, offering efficient compute times. Neural flows (Biloš et al., 2021) learn the solution path of the ODE directly, bypassing the need for a solver. Attention-based methods like multi-time attention networks (MANs) (Shukla & Marlin, 2021) use specialized embedding modules to map irregular and sparse observations to continuous-time representations. Graph-based methods such as GraFITi (Yalavarthi et al., 2024) convert irregularly sampled time series into sparse structure graphs, reformulating forecasting as edge weight prediction using graph neural networks (GNNs). Transformable patching graph neural networks (T-PATCHGNN) (Zhang et al., 2024) transform each univariate irregular time series into transformable patches with uniform temporal resolution.

Spectral methods have been explored to capture periodic behaviors in time series. TimesNet (Wu et al., 2023) and other Fourier-based models (Alaa et al., 2021; Crabbé et al., 2024) leverage the Discrete Fourier Transform (DFT) to extract frequency components, improving long-term forecasting and imputation. Additionally, Fons et al. (2022; 2024) introduced a *Fourier loss* to better preserve frequency structure in learning, aligning time-domain reconstructions with their spectral representations. However, like other FFT-based methods, these approaches still require uniform sampling or pre-interpolation and thus can distort frequency estimates under high missingness or irregular sampling (VanderPlas, 2018). By contrast, the Lomb–Scargle periodogram (Lomb, 1976; Scargle, 1982) computes frequency spectra directly from unevenly sampled data, sidestepping the need for interpolation. Despite its advantages, Lomb–Scargle remains underutilized in machine learning, with most of its applications still confined to astronomy and signal processing.

While prior work has explored both time-domain diffusion-based imputation (Tashiro et al., 2021) and frequency-based representations (Wu et al., 2023; Yang et al., 2024a), our approach is the first to integrate a differentiable Lomb–Scargle estimator into the diffusion process. By conditioning time-domain diffusion on Lomb–Scargle-derived spectral information, we avoid spectral distortions introduced by uniform resampling. This enables more faithful recovery of underly-

ing frequency structures, ensuring that imputed time-series data align with their true spectral content.

# 3. Background and Motivation

In this section, we present the mathematical foundations of *Lomb–Scargle* (Lomb, 1976; Scargle, 1982) and demonstrate its effectiveness as a robust alternative to traditional FFT-based frequency analysis when handling missing data. Additionally, in Sec. 3.3, we summarize the conditional diffusion framework for time series imputation introduced by Tashiro et al. (2021).

## 3.1. FFT vs Lomb–Scargle

Time series data are frequently subject to missing or irregularly spaced observations, caused by sensor failures, asynchronous measurements, or domain-specific constraints (Yang et al., 2024b). Although the Fast Fourier Transform (FFT) is a powerful tool for frequency analysis, it implicitly assumes *uniform sampling*. When data points are missing, common approaches include using interpolation or zero-filling to address the missing observations. Both strategies can distort the true frequency content, leading to spurious or attenuated peaks (VanderPlas, 2018). This is especially problematic at high missingness levels, where interpolation becomes highly unreliable.

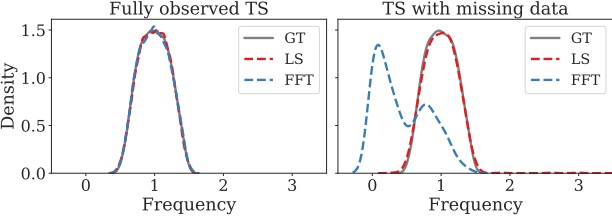

Figure 1: Density of leading frequencies for FFT and Lomb-Scargle (LS) on a synthetic sine dataset. (Left) Fully observed time series. (Right) Time series with 75% missing data. The interpolation required by FFT significantly distorts the spectral distribution, whereas LS better preserves the original frequency structure.

This is illustrated in Figure 1, where we show a comparison of FFT vs Lomb–Scargle applied to a synthetic dataset of sine waves with *known frequencies*. Under *fully observed* conditions, both FFT and Lomb–Scargle reliably recover the ground-truth leading frequencies. However, once we artificially remove data (75% missing at random), the FFT with linear interpolation yields shifted or spurious peaks, whereas Lomb–Scargle remains well-aligned with the ground truth frequencies.

## 3.2. Lomb–Scargle Periodogram

Let $S = \{s_1, \ldots, s_L\}$ denote a set of time indices and $\boldsymbol{x} = [x_{s_1}, \ldots, x_{s_L}]^{\mathsf{T}} \in \mathbb{R}^L$ denote the observed values defining a potentially irregular time series. We define a frequency grid $\boldsymbol{\omega} = [\omega_1, \ldots, \omega_J]$ of interest, where each $\omega_j = 2\pi f_j$ denotes the corresponding angular frequency (in radians) of the $j$th frequency component $f_j$. For each frequency component $\omega$, Lomb–Scargle approximates the *power spectral density* (PSD) $P(\omega)$ by fitting sinusoids directly to the *observed* data, circumventing the need to fill in missing points:

$$P(\omega) = \frac{\left(\sum_i [x_{s_i} - \bar{x}] \cos[\omega \phi_i]\right)^2}{\sum_i \cos^2[\omega \phi_i]} + \frac{\left(\sum_i [x_{s_i} - \bar{x}] \sin[\omega \phi_i]\right)^2}{\sum_i \sin^2[\omega \phi_i]},$$

with $\bar{x} = \frac{1}{L} \sum_{i=1}^{L} x_{s_i}$ denotes the sample mean of the observed values, and $\phi_i = s_i - \tau$, where $\tau$ is a time shift introduced to ensure the periodogram is invariant to time translations defined as:

$$\tau = \frac{1}{2\omega} \tan^{-1} \left( \frac{\sum_i \sin(2\omega s_i)}{\sum_i \cos(2\omega s_i)} \right).$$

This shift aligns the cosine and sine terms optimally with the observed data, ensuring robust spectral estimates. Unlike the FFT, no uniform grid or zero-filling is required, making Lomb–Scargle particularly effective for irregularly sampled time series. We provide a basic review on the derivation of the Lomb–Scargle periodogram in Appendix A.

**False Alarm Probability.** It can be shown that under the assumption of additive standard Gaussian noise, $P(\omega)$ follows a $\chi^2$ distribution with 2 degrees of freedom. This fact can be exploited to approximate the false alarm probability (FAP) for each frequency component $\omega_k$, denoted by $\mathbb{P}_{FA}(\omega_k)$, using the following expression:

$$\mathbb{P}_{FA}(\omega) = 1 - [1 - \exp(-P_f(\omega))]^{J_{\text{eff}}}, \qquad (1)$$

where $J_{\text{eff}}$ denotes an effective number of independent frequencies, which can be estimated using heuristics in practice. The FAP can be used to detect and filter out spurious frequency components in the periodogram. Using the FAP, we can define the weighting function as:

$$w(\omega_k) = \frac{1}{\mathbb{P}_{FA}(\omega_k) + \epsilon}, \qquad (2)$$

where $\epsilon$ is a small constant to prevent division by zero. This weighting scheme ensures that frequencies with lower FAP (i.e., more significant frequencies) have a greater impact on the loss.

## 3.3. Conditional Diffusion for Time Series Imputation

We summarize the conditional diffusion framework for time series imputation (Tashiro et al., 2021). Let us consider

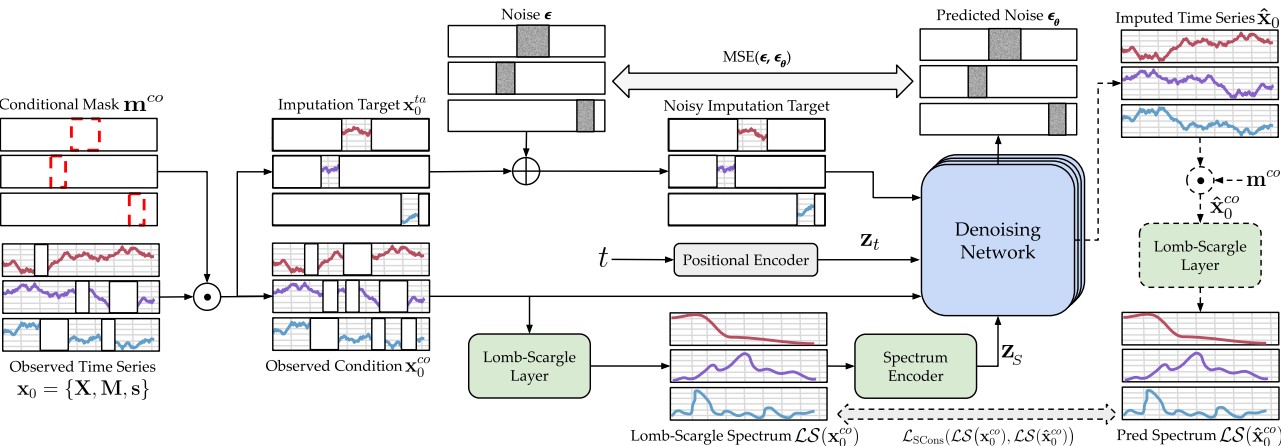

Figure 2: Diagram of our *Lomb–Scargle Conditioned Diffusion* (LSCD) approach for time series imputation.

$N$ multivariate time series, each denoted by $\{\mathbf{X}, \mathbf{M}, \mathbf{S}\}$, where $\mathbf{X} \in \mathbb{R}^{K \times L}$ corresponds to the time series values with $K$ features over $L$ time steps, and $\mathbf{s} = \{s_1, \ldots, s_L\}$ contains the timestamps. We assume that some entries of $\mathbf{X}$ are missing, as defined by the observation mask $\mathbf{M} \in \{0,1\}^{K \times L}$. Our goal is to model and then sample from the *conditional distribution* $p_\theta(\mathbf{x}_0^{ta} \mid \mathbf{x}_0^{co})$, where $\mathbf{x}_0^{co} = \mathbf{M} \odot \mathbf{X}$ corresponds to the conditional observations, and $\mathbf{x}_0^{ta} = (1 - \mathbf{M}) \odot \mathbf{X}$ to the imputation targets, i.e., the missing entries that need to be imputed, with $\odot$ denoting element-wise multiplication.

**Forward Process.** To learn this conditional distribution using a *score-based diffusion model*, we adopt a forward process that gradually adds noise to the *target* portion $\mathbf{x}_0^{ta}$. Specifically, for diffusion steps $t = 1, 2, \ldots, T$,

$$q(\mathbf{x}_t^{ta} \mid \mathbf{x}_{t-1}^{ta}) = \mathcal{N}\left(\sqrt{1 - \beta_t}\mathbf{x}_{t-1}^{ta}, \beta_t \mathbf{I}\right),$$

where $\{\beta_t\}$ is a noise schedule. Under suitable definitions of $\alpha_t := \prod_{i=1}^{t}(1 - \beta_i)$, we have the closed-form

$$\mathbf{x}_t^{ta} = \sqrt{\alpha_t}\,\mathbf{x}_0^{ta} + \sqrt{1 - \alpha_t}\,\boldsymbol{\epsilon}, \quad \boldsymbol{\epsilon} \sim \mathcal{N}(\mathbf{0}, \mathbf{I}).$$

**Reverse Process.** The *reverse process*, parameterized by $\theta$, attempts to denoise $\mathbf{x}_T^{ta}$ step by step back to $\mathbf{x}_0^{ta}$, while being conditioned on $\mathbf{x}_0^{co}$. Hence we define

$$p_\theta(\mathbf{x}_0^{ta}, \ldots, \mathbf{x}_T^{ta} \mid \mathbf{x}_0^{co}) = p(\mathbf{x}_T^{ta}) \prod_{t=1}^{T} p_\theta(\mathbf{x}_{t-1}^{ta} \mid \mathbf{x}_t^{ta}, \mathbf{x}_0^{co}),$$

where

$$p_\theta(\mathbf{x}_{t-1}^{ta} \mid \mathbf{x}_t^{ta}, \mathbf{x}_0^{co}) = \mathcal{N}\left(\mathbf{x}_{t-1}^{ta}; \boldsymbol{\mu}_\theta(\mathbf{x}_t^{ta}, t \mid \mathbf{x}_0^{co}), \sigma^2(t)\mathbf{I}\right).$$

In practice (Ho et al., 2020; Song et al., 2021), $\boldsymbol{\mu}_\theta$ is parameterized via a denoising network $\boldsymbol{\epsilon}_\theta(\mathbf{x}_t^{ta}, t \mid \mathbf{x}_0^{co})$:

$$\boldsymbol{\mu}_\theta(\mathbf{x}_t^{ta}, t \mid \mathbf{x}_t^{co}) = \frac{1}{\alpha_t}\left(\mathbf{x}_t^{ta} - \frac{\beta_t}{\sqrt{1 - \alpha_t}}\boldsymbol{\epsilon}_\theta(\mathbf{x}_t^{ta}, t \mid \mathbf{x}_t^{co})\right)$$

$$\sigma^2(t) = \begin{cases} \frac{1 - \alpha_{t-1}}{1 - \alpha_t}\beta_t & t > 1 \\ \beta_1 & t = 1 \end{cases}$$

**Training.** The parameters $\theta$ are learned by matching the predicted noise $\boldsymbol{\epsilon}_\theta$ to the true noise $\boldsymbol{\epsilon}$. We minimize

$$\mathcal{L}(\theta) = \mathbb{E}\left[\|\boldsymbol{\epsilon} - \boldsymbol{\epsilon}_\theta(\mathbf{x}_t^{ta}, t \mid \mathbf{x}_0^{co})\|^2\right],$$

where $\mathbf{x}_t^{ta}$ is sampled from the forward process. In practice, during training a *conditional mask* $\mathbf{m}^{co} \in \{0,1\}^{K \times L}$ is introduced to artificially split the observed values into $\mathbf{x}_0^{co} = \mathbf{m}^{co} \odot \mathbf{X}$ and $\mathbf{x}_0^{ta} = (\mathbf{M} - \mathbf{m}^{co}) \odot \mathbf{X}$, in order to train the conditional denoising function. At inference time (imputation), we generate samples of $\mathbf{x}_0^{ta}$ by initializing $\mathbf{x}_T^{ta}$ with random Gaussian noise and iteratively applying the learned reverse transitions, while conditioning on the observed data $\mathbf{x}_0^{co}$.

## 4. Methodology: Spectrum-Conditioned Diffusion

We now present our *Lomb–Scargle Conditioned Diffusion* (LSCD) framework for time series imputation, illustrated in Fig. 2. Building on the conditional diffusion approach in Sec. 3.3, we introduce a Lomb–Scargle layer that supplies two crucial forms of frequency-domain supervision. At each denoising step, a Lomb–Scargle-based spectral representation is learned and provided as conditional information. Furthermore, after the main diffusion training, we incorporate a spectral alignment term that encourages the final imputed time series to match the Lomb–Scargle spectrum extracted from the observed data.

## 4.1. Time-Domain Diffusion with Lomb–Scargle Conditioning

Given a partially observed time series $\mathbf{x}_0$ defined by $\{\mathbf{X}, \mathbf{M}, \mathbf{S}\}$, we divide the observed values in two parts, according to a randomly generated *conditional mask* $\boldsymbol{m}^{co}$:

$$\mathbf{x}_0^{co} := \mathbf{m}^{co} \odot \mathbf{X} \qquad \mathbf{x}_0^{ta} := (\mathbf{M} - \mathbf{m}^{co}) \odot \mathbf{X}$$

thus defining an *imputation target* $\mathbf{x}_0^{ta}$ (i.e. artificially generated missing entries) and an *observation condition* $\mathbf{x}_0^{co}$ of observed values. Following Sec. 3.2, we define $\mathcal{LS}(\mathbf{x}_0^{co}) \in \mathbb{R}^J$ as the Lomb-Scargle periodogram of $\mathbf{x}_0^{co}$, evaluated over a frequency grid $\boldsymbol{\omega} = \{\omega_1, \ldots, \omega_J\}$. The spectrum is filtered via *False Alarm Probability* (FAP), as detailed in Eq. 1, and we additionally apply a log transform and normalization.

**Forward Process.** We follow the usual diffusion setup, wherein at each diffusion step $t = 1, \ldots, T$ the *imputation target* $\mathbf{x}_{t-1}^{ta}$ is perturbed according to

$$\mathbf{x}_t^{ta} = \sqrt{\alpha_t}\,\mathbf{x}_{t-1}^{ta} + \sqrt{1 - \alpha_t}\,\boldsymbol{\epsilon}_t, \quad \boldsymbol{\epsilon}_t \sim \mathcal{N}(\mathbf{0}, \mathbf{I}),$$

with $\{\alpha_t\}$ a noise schedule.

**Backward (Denoising) Process.** A denoising network $\boldsymbol{\epsilon}_\theta(\mathbf{x}_t^{ta}, t \mid \mathbf{x}_0^{co}, \mathcal{LS}(\mathbf{x}_0^{co}))$ is trained to invert the forward noising, predicting the noise $\boldsymbol{\epsilon}_t$ given both the noisy time-domain data $\mathbf{x}_t^{ta}$ and the fixed spectral representation $\mathcal{LS}(\mathbf{x}_0^{co})$. The final sampling procedure iterates the reverse transitions $\mathbf{x}_t^{ta} \longmapsto \mathbf{x}_{t-1}^{ta}$, conditioned on $\mathbf{x}_0^{co}$ and $\mathcal{LS}(\mathbf{x}_0^{co})$, until an imputed $\mathbf{x}_0^{ta}$ is obtained.

## 4.2. Attention-Based Spectrum Encoder

Our approach leverages all frequency components from the spectrum as a conditioning signal for diffusion. To that end, we incorporate a *spectral encoder*:

$$\mathbf{z}_S = \mathcal{E}_{\text{spec}}(\mathcal{LS}(\mathbf{x}_0^{co})),$$

composed of two multi-head self-attention layers, designed to encode *inter-frequency* and *inter-feature* dependencies. The resulting embedded representation of the spectrum $\mathbf{z}_S$ is then incorporated into each step of the denoising process. This allows the network to exploit rich frequency-domain cues while reconstructing the time-domain signal. We refer the reader to Appendix E.2 for additional details on the encoder and denoising network architectures.

## 4.3. Spectral Consistency Loss

To reinforce alignment between the final imputed time series and the observed spectrum, we perform a final refinement phase during training, incorporating a *spectral consistency loss*:

$$\mathcal{L}_{\text{SCons}} = \|\mathcal{LS}(\mathbf{x}_0^{co}) - \mathcal{LS}(\widehat{\mathbf{x}}_0^{co})\|_2^2,$$

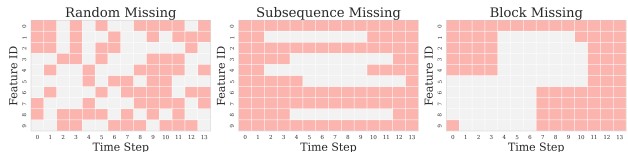

Figure 3: Visualization of the three missingness mechanisms used in our study.

with $\widehat{\mathbf{x}}_0^{co} = \widehat{\mathbf{x}}_0 \odot \mathbf{m}^{co}$, where $\widehat{\mathbf{x}}_0$ is the fully reconstructed time series (i.e. observed plus imputed parts), obtained by following the backward denoising process from $t = T$ to $t = 0$. The pipeline to compute $\mathcal{L}_{\text{SCons}}$ is illustrated in dashed lines in Fig. 2. This term penalizes discrepancies in the Lomb–Scargle periodograms, ensuring that the learned reconstruction not only fits the observed data in the time domain, but also preserves the essential frequency structure.

# 5. Experiments

In this section, we evaluate LSCD on both synthetic and real-world datasets. We focus on time series imputation under varying degrees of missingness (e.g., 10%, 50%, 90% missing data). We also include ablation studies to isolate the impact of key components such as Lomb–Scargle conditioning, the spectral encoder $\mathcal{E}_{\text{spec}}$, and the spectral consistency loss $\mathcal{L}_{\text{SCons}}$. We outline the datasets, baselines, metrics, and ablation protocols below.

## 5.1. Experimental Setup

### 5.1.1. DATASETS

**Synthetic Sine Waves.** We create a controlled dataset of sine waves with known frequencies $(f_1, f_2, \ldots)$ and various amplitudes. To make the setting more realistic, the dataset is initially generated with 10% missing data. We evaluate our method using three missingness mechanisms: *MCAR*, *sequence missing*, and *block missing*. *MCAR* introduces missing values randomly, independent of the data itself, serving as a baseline. *Sequence missing* creates contiguous gaps in time, simulating sensor downtimes or transmission failures. *Block missing* removes entire sections of data, mimicking large-scale outages. These mechanisms test the model's ability to handle both short-term and long-term dependencies. Each missingness mechanism is applied at three levels (10%, 50%, and 100%) on top of the initial missing data to assess the robustness of our method under increasing data loss. Figure 3 visualizes these missingness patterns, illustrating their real-world relevance, from random noise to structured data loss. Additional details on dataset generation and missingness parameters can be found in Appendix D.

**Real datasets** We conduct experiments on two real-world datasets with missing data. The first dataset, PhysioNet (Silva et al., 2012), comprises 4,000 health measurements from ICU patients, covering 35 features. Following the standard pre-processing procedure (Cao et al., 2018; Tashiro et al., 2021), each measurement is processed hourly, resulting in time series with 48 time steps. The preprocessed dataset contains approximately 80% missing values. As there is no ground truth for missing data, we randomly select 10%, 50%, and 90% of the observed values to serve as ground truth for the test set. The second dataset consists of PM2.5 air quality measurements collected from 36 stations in Beijing over a 12-month period (Yi et al., 2016). Following previous work (Cao et al., 2018; Tashiro et al., 2021), we construct time series with 36 consecutive time steps. This dataset exhibits approximately 13% missing data, with non-random missingness patterns and an artificial ground truth containing structured missingness.

### 5.1.2. BASELINES

We compare our model against different imputation methods, including classical statistic-based methods Mean and Lerp, deep learning based BRITS (Cao et al., 2018), GP-VAE (Fortuin et al., 2020), US-GAN (Miao et al., 2021), TimesNet (Wu et al., 2023), SAITS (Du et al., 2023), CSDI (Tashiro et al., 2021) and a foundation model for time series, ModernTCN (Donghao & Xue, 2024).

### 5.2. Evaluation Metrics

We evaluate the performance of our models using a combination of time-domain and frequency-domain metrics to comprehensively assess both the accuracy of imputed values and the preservation of the underlying spectrum. In the time domain, we utilize *Mean Absolute Error* (**MAE**) and *Root Mean Squared Error* (**RMSE**) to quantify the accuracy of the imputed values, providing measures of both average absolute deviation and squared error sensitivity to larger discrepancies. To assess the fidelity of our models in capturing the spectral properties of the time series, we calculate the *Spectral Mean Absolute Error* (**S-MAE**) between the estimated *Power Spectral Density* (PSD) of the imputed signal and the ground-truth PSD. The spectrum is computed using both observed (conditional) and imputed values, while missing values in the original time series are masked to ensure a fair comparison. To focus on relative differences in spectral shape, we normalize both PSDs so their total power sums to one before computing **S-MAE** as the mean absolute difference between them. This metric quantifies how well the model preserves the frequency characteristics of the time series.

### 5.3. Results on Synthetic Sine Dataset

Table 1 presents the imputation performance of various methods on the Sines dataset, across different missingness mechanisms and levels. Our proposed method consistently achieves the best overall performance, particularly excelling in spectral reconstruction (S-MAE). For MCAR data, our method significantly outperforms CSDI for all levels of missingness, showcasing the effectiveness of our approach in handling random data loss. Notably, CSDI consistently generates results that have good agreement in S-MAE, particularly in sequence and block missing cases, indicating that its probabilistic modeling helps retain spectral properties. SAITS demonstrates competitive performance in time-domain imputation (MAE) for some cases (e.g., sequence missing at 10%), but consistently exhibits poor spectral reconstruction (S-MAE). This highlights that SAITS, while effective in the time domain, fails to capture the underlying frequency structure of the data.

Figure 4 evaluates the spectral consistency of different models on the Sines dataset with 50% missing data at random. The top row compares the distribution of leading frequencies between the ground truth (GT) and predicted (Pred) time series. Our model closely follows the GT distribution, while SAITS and TimesNet exhibit broader spreads, and TimesNet shows more pronounced deviations, particularly around peaks at higher frequencies. This suggests that our model better preserves the dominant spectral characteristics. The bottom row presents the PSD difference $PSD(GT) - PSD(Pred)$, where the black line represents the mean difference, and the shaded area indicates one standard deviation across samples. We observe that our model maintains a difference centered around zero with minimal variance, indicating accurate spectral reconstruction. In contrast, SAITS and CSDI exhibit larger variability, suggesting inconsistent frequency estimation, while TimesNet shows a systematic bias, particularly in higher frequency ranges. These results further exhibit the effectiveness of our method in preserving the spectral properties of time series, ensuring more stable and reliable frequency reconstruction.

### 5.4. Results on Real Data

To assess the performance of our spectrum-conditioned diffusion approach, we conduct experiments on two diverse real-world time-series datasets—Physionet and PM2.5—under multiple missing-data scenarios. Table 2 summarizes the results, where we compare against a wide range of baselines. We repeat experiments five times and report the averaged result. We first consider Physionet with three different missing rates: 10%, 50%, and 90%. Across all these scenarios, our method consistently achieves the best or near-best time-domain results (MAE and RMSE) while also exhibiting competitive S-MAE scores. Notably,

Table 1: Imputation performance of time series reconstruction and frequency spectrum over sines datasets.

| TYPE | % | METRIC | MEAN | LERP | BRITS | GPVAE | US-GAN | TIMESNET | CSDI | SAITS | MODERNTCN | OURS |
|------|-----|--------|-------|-------|-------|-------|--------|----------|-------|-------|-----------|------|
| POINT | 0.1 | MAE | 1.380 | 1.305 | 0.943 | 1.399 | 0.933 | 1.220 | 1.336 | 0.885 | 0.973 | **0.765** |
| | | RMSE | 1.947 | 1.991 | 1.657 | 1.986 | 1.636 | 1.803 | 1.889 | 1.569 | 1.727 | **1.453** |
| | | S-MAE | 0.081 | 0.081 | 0.052 | 0.082 | 0.053 | 0.069 | 0.008 | 0.043 | 0.049 | **0.003** |
| | 0.5 | MAE | 1.373 | 1.449 | 1.095 | 1.383 | 1.152 | 1.481 | 1.359 | 1.041 | 1.129 | **0.975** |
| | | RMSE | 1.930 | 2.070 | 1.759 | 1.950 | 1.845 | 2.017 | 1.922 | 1.699 | 1.817 | **1.658** |
| | | S-MAE | 0.264 | 0.324 | 0.170 | 0.266 | 0.191 | 0.239 | 0.027 | 0.159 | 0.173 | **0.014** |
| | 0.9 | MAE | 1.375 | 1.586 | 1.320 | 1.377 | 1.369 | 1.579 | 1.361 | 1.292 | 1.360 | **1.271** |
| | | RMSE | 1.935 | 2.142 | 1.899 | 1.938 | 1.970 | 2.143 | 1.925 | 1.878 | 1.963 | **1.870** |
| | | S-MAE | 0.439 | 0.572 | 0.383 | 0.439 | 0.407 | 0.462 | 0.044 | 0.375 | 0.406 | **0.036** |
| SEQ | 0.1 | MAE | 1.353 | 1.542 | 1.330 | 1.355 | 1.384 | 1.391 | 1.413 | **1.323** | 1.329 | 1.359 |
| | | RMSE | 1.905 | 2.092 | 1.915 | 1.908 | 1.995 | 1.959 | 1.988 | **1.890** | 1.931 | 1.962 |
| | | S-MAE | 0.055 | 0.075 | 0.056 | 0.054 | 0.061 | 0.062 | 0.006 | 0.055 | 0.056 | **0.005** |
| | 0.5 | MAE | 1.374 | 1.564 | 1.347 | 1.376 | 1.393 | 1.467 | 1.378 | 1.342 | 1.354 | **1.316** |
| | | RMSE | 1.934 | 2.115 | 1.928 | 1.936 | 1.999 | 2.038 | 1.943 | 1.917 | 1.960 | **1.913** |
| | | S-MAE | 0.271 | 0.369 | 0.269 | 0.271 | 0.297 | 0.321 | 0.028 | 0.268 | 0.277 | **0.026** |
| | 0.9 | MAE | 1.386 | 1.573 | 1.362 | 1.388 | 1.403 | 1.489 | 1.372 | 1.352 | 1.375 | **1.313** |
| | | RMSE | 1.946 | 2.127 | 1.941 | 1.949 | 2.007 | 2.062 | 1.943 | 1.929 | 1.982 | **1.913** |
| | | S-MAE | 0.288 | 0.389 | 0.286 | 0.288 | 0.305 | 0.338 | 0.029 | 0.283 | 0.292 | **0.027** |
| BLOCK | 0.1 | MAE | 1.306 | 1.507 | 1.255 | 1.309 | 1.334 | 1.379 | 1.304 | 1.268 | 1.275 | **1.259** |
| | | RMSE | 1.807 | 2.014 | 1.786 | 1.811 | 1.885 | 1.898 | 1.804 | 1.785 | 1.825 | **1.774** |
| | | S-MAE | 0.105 | 0.146 | 0.100 | 0.104 | 0.116 | 0.124 | 0.011 | 0.103 | 0.106 | **0.010** |
| | 0.5 | MAE | 1.306 | 1.505 | 1.279 | 1.308 | 1.333 | 1.451 | 1.314 | 1.285 | 1.309 | **1.269** |
| | | RMSE | 1.815 | 2.014 | 1.806 | 1.817 | 1.881 | 1.978 | 1.835 | **1.804** | 1.852 | 1.810 |
| | | S-MAE | 0.287 | 0.383 | 0.278 | 0.286 | 0.306 | 0.344 | 0.029 | 0.285 | 0.296 | **0.027** |
| | 0.9 | MAE | 1.339 | 1.523 | 1.319 | 1.340 | 1.359 | 1.506 | 1.329 | **1.320** | 1.356 | 1.320 |
| | | RMSE | 1.868 | 2.052 | 1.862 | 1.869 | 1.927 | 2.054 | 1.874 | **1.859** | 1.916 | 1.870 |
| | | S-MAE | 0.359 | 0.473 | 0.351 | 0.358 | 0.376 | 0.439 | 0.036 | 0.358 | 0.374 | **0.035** |

Figure 4: (Top) Distribution of leading frequency on the Sines dataset (Missing Point 50%). (Bottom) Difference between the power spectral densities (PSD) of the ground truth and predictions, with the black line representing the mean difference and the shaded area indicating one standard deviation. Our approach shows the best performance.

at the lower 10% missing rate, our approach yields the lowest MAE and RMSE, improving upon the already strong baseline CSDI. For the spectral metric (S-MAE), we again outperform all alternatives, demonstrating more accurate frequency-domain reconstruction. At 50% and 90% missing rates, our approach maintains the lowest scores in all

metrics, with S-MAE sometimes on par with CSDI. These results suggest that explicit conditioning on frequency components helps maintain signal characteristics even under high data sparsity. In the PM2.5 pollution dataset, our model reports significant gains in MAE and RMSE compared to the baselines. Additionally, we also observe a lower S-MAE

Table 2: Imputation performance of time series reconstruction and frequency spectrum over real datasets.

| DATASET | MISS | METRIC | MEAN | LERP | BRITS | GPVAE | US-GAN | TIMESNET | CSDI | SAITS | MODERNTCN | OURS |
|---|---|---|---|---|---|---|---|---|---|---|---|---|
| PHYSIONET | 10% | MAE | 0.714 | 0.372 | 0.278 | 0.469 | 0.323 | 0.375 | 0.219 | 0.232 | 0.351 | **0.211** |
| | | RMSE | 1.035 | 0.708 | 0.693 | 0.783 | 0.662 | 0.690 | 0.545 | 0.583 | 0.697 | **0.494** |
| | | S-MAE | 0.032 | 0.020 | 0.016 | 0.026 | 0.020 | 0.022 | 0.013 | 0.014 | 0.020 | **0.012** |
| | 50% | MAE | 0.711 | 0.417 | 0.385 | 0.521 | 0.449 | 0.453 | 0.307 | 0.315 | 0.440 | **0.303** |
| | | RMSE | 1.091 | 0.840 | 0.833 | 0.907 | 0.852 | 0.840 | 0.672 | 0.735 | 0.803 | **0.664** |
| | | S-MAE | 0.111 | 0.087 | 0.064 | 0.083 | 0.076 | 0.076 | **0.052** | 0.055 | 0.071 | **0.052** |
| | 90% | MAE | 0.710 | 0.565 | 0.560 | 0.642 | 0.670 | 0.642 | 0.481 | 0.565 | 0.647 | **0.479** |
| | | RMSE | 1.097 | 0.993 | 0.975 | 1.038 | 1.060 | 1.031 | 0.834 | 0.971 | 1.026 | **0.832** |
| | | S-MAE | 0.148 | 0.189 | 0.104 | 0.124 | 0.125 | 0.131 | **0.093** | 0.108 | 0.137 | **0.093** |
| PM25 | 10% | MAE | 50.685 | 15.363 | 16.519 | 23.941 | 32.999 | 22.685 | 9.670 | 15.424 | 24.089 | **9.069** |
| | | RMSE | 66.558 | 27.658 | 26.775 | 40.586 | 48.951 | 39.336 | 19.093 | 30.558 | 40.052 | **17.914** |
| | | S-MAE | 0.135 | 0.039 | 0.039 | 0.060 | 0.080 | 0.056 | 0.023 | 0.034 | 0.059 | **0.022** |

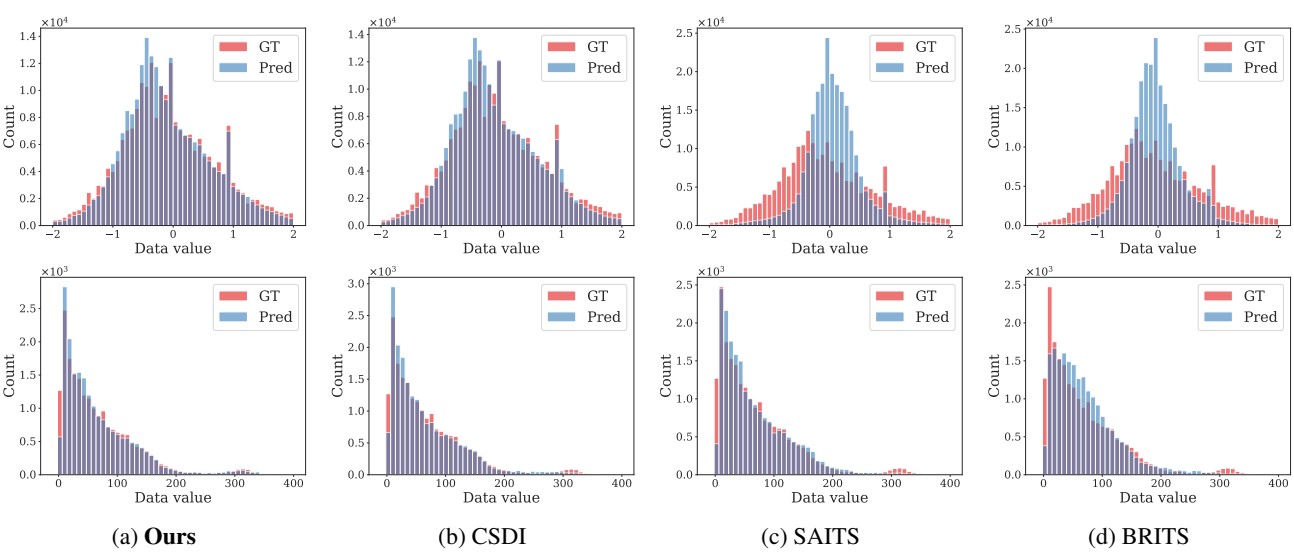

(a) **Ours**     (b) CSDI     (c) SAITS     (d) BRITS

Figure 5: Distribution of imputed values across different models on real datasets. (Top) Physionet dataset with 10% missing data; (Bottom) PM25 dataset.

than competing methods, indicating stronger preservation of the signal's spectral structure. Finally, Figure 5 shows a comparison between the imputed values and the observed ones for Physionet (top) and PM25 (bottom). In Physionet, CSDI and our method capture the general distribution shape, but LSCD has a slight advantage for larger values, while SAITS and BRITS show a concentration in lower values, without covering the tails of the distribution. For PM25, both CSDI and our method show similar performance, but our method covers more accurately larger values. These experiments underscore the advantage of using Lomb-Scargle-based spectrum conditioning within a diffusion framework. By circumventing the need to fill in missing points before spectral analysis, our approach better preserves the time series' inherent frequency characteristics. As evidenced by strong results on both Physionet and PM2.5, frequency-aware conditioning leads not only to lower time-domain errors but also to more accurate frequency reconstructions.

### 5.5. Ablation Studies

To validate the contribution of each component in our LSCD framework, we conduct ablation studies on the Physionet and PM25 datasets, progressively removing key elements. Results are summarized in Table 3. The full model achieves the best performance across all settings, demonstrating the effectiveness of Lomb-Scargle conditioning ($\mathcal{LS}$), spectral encoder ($\mathcal{E}_{\text{spec}}$), and spectral consistency loss ($\mathcal{L}_{\text{SCons}}$). Removing $\mathcal{L}_{\text{SCons}}$ marginally increases MAE, highlighting its role in aligning imputations with spectral properties. Further removing $\mathcal{E}_{\text{spec}}$ degrades performance more significantly, emphasizing the importance of encoding inter-frequency dependencies. Removing $\mathcal{LS}$ conditioning entirely causes the largest degradation, underscoring the necessity of direct spectral supervision. Notably, S-MAE remains stable at high missing rates. These results confirm that all components contribute to accurate time-domain imputation while preserving spectral fidelity.

Table 3: Ablation of LSCD in Physionet and PM2.5.

| METHOD | 10% MISS. | | 50% MISS. | | 90% MISS. | | PM25 | |
|---|---|---|---|---|---|---|---|---|
| | MAE | S-MAE | MAE | S-MAE | MAE | S-MAE | MAE | S-MAE |
| OURS | **0.211** | **0.012** | **0.303** | **0.052** | **0.479** | **0.093** | **9.069** | **0.0219** |
| w/o $\mathcal{L}_{SCons}$ | 0.213 | **0.012** | **0.303** | **0.052** | 0.480 | **0.093** | 9.085 | 0.0223 |
| w/o $\mathcal{L}_{SCons}, \mathcal{E}_{\text{SPEC}}$ | 0.218 | 0.013 | 0.304 | **0.052** | 0.480 | **0.093** | 9.334 | 0.0222 |
| w/o $\mathcal{L}_{SCons}, \mathcal{E}_{\text{SPEC}}, \mathcal{LS}$ | 0.219 | 0.013 | 0.307 | **0.052** | 0.481 | **0.093** | 9.669 | 0.0239 |

## 6. Limitations

Lomb–Scargle natively supports irregularly sampled data. However, our LSCD architecture for time series imputation works with grid-based data and thus our differentiable implementation of Lomb–Scargle is adapted for this type of input. While score-based diffusion models such as CSDI (Tashiro et al., 2021) and LSCD rely on a fixed time grid, they are able to operate on irregularly sampled time series, *provided that the interpolation time points are known at training time*. In contrast, continuous-time methods only require knowledge of the interpolation time points at inference time.

## 7. Conclusions

We presented *Lomb–Scargle Conditioned Diffusion* (LSCD), a novel approach for time series imputation that leverages a diffusion model conditioned on the Lomb–Scargle periodogram. Our method directly incorporates spectral information within the diffusion framework, ensuring that both time-domain and frequency-domain characteristics of the data are preserved.

Our experimental evaluation on synthetic and real-world datasets demonstrated that LSCD consistently outperforms existing imputation methods in both time-domain accuracy (MAE, RMSE) and spectral consistency (S-MAE). Notably, our method maintains superior performance even under extreme missingness (up to 90%), demonstrating its robustness in practical scenarios.

Our approach employs a *spectral encoder* to process the Lomb–Scargle spectrum and a *spectral consistency loss* to reinforce alignment between imputed signals and their frequency representations. Our ablation studies confirmed that each of these components contributes significantly to the overall effectiveness of our method. The spectral encoder captures inter-frequency and inter-feature dependencies, enhancing the model's ability to utilize spectral cues, while the spectral consistency loss ensures that frequency components are faithfully reconstructed in the imputation.

Beyond imputation, we see significant potential for Lomb–Scargle in machine learning applications involving missing or irregular data. To facilitate broader adoption, we provide a differentiable implementation which can be seamlessly integrated into learning pipelines. We hope this will encourage further exploration of spectral-guided learning methods.

## Impact Statement

Our work focuses on improving time series imputation. Potential applications include healthcare (e.g., filling gaps in patient vitals), climate modeling (handling sparse sensor readings), and finance (dealing with stock transactions). By restoring continuity and enhancing frequency fidelity, we enable more reliable downstream analyses, which can lead to better decision-making in critical domains.

## Acknowledgments

This paper was prepared for informational purposes in part by the Artificial Intelligence Research group of JPMorgan Chase & Co. and its affiliates ("JP Morgan") and is not a product of the Research Department of JP Morgan. JP Morgan makes no representation and warranty whatsoever and disclaims all liability, for the completeness, accuracy or reliability of the information contained herein. This document is not intended as investment research or investment advice, or a recommendation, offer or solicitation for the purchase or sale of any security, financial instrument, financial product or service, or to be used in any way for evaluating the merits of participating in any transaction, and shall not constitute a solicitation under any jurisdiction or to any person, if such solicitation under such jurisdiction or to such person would be unlawful.

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

## A. Lomb-Scargle Discussion

The Lomb-Scargle periodogram provides a spectral representation of a signal. It is typically used to understand the power spectrum of an irregularly sampled time series sample, typically in the context of astronomy data. Consider a univariate signal $\mathbf{x} = (x_{t_1}, \ldots, x_{t_N})$, which is assumed to be irregularly sampled at times $t_1, \ldots, t_N$. For a given frequency $f$, the idea behind the Lomb-Scargle periodogram is to fit $\mathbf{x}$ to a sinusoid using least-squares. Equivalently, we assume that at each time $t_n$, the signal can be modeled as a noisy sinusoid:

$$x_{t_n} = A\cos(2\pi f t_n + \phi) + \epsilon_{t_n} \tag{3}$$

where $\epsilon_t \sim \mathcal{N}(0, \sigma^2)$ is assumed to be white Gaussian noise. Using a trigonometric identity, the model can equivalently be represented as:

$$x_{t_n} = \alpha_1 \cos(2\pi f t_n) + \alpha_2 \sin(2\pi f t_n) + \epsilon_{t_n} \tag{4}$$

where the amplitude $A$ and phase $\phi$ can be related to the parameters $\alpha_1$ and $\alpha_2$ as follows:

$$A = \sqrt{\alpha_1^2 + \alpha_2^2} \tag{5}$$

$$\phi = \tan^{-1}\left(-\frac{\alpha_2}{\alpha_1}\right) \tag{6}$$

The model formulation in (4) is convenient, because it can be expressed as a linear model with additive Gaussian noise:

$$\underbrace{\begin{bmatrix} x_{t_1} \\ \vdots \\ x_{t_N} \end{bmatrix}}_{\mathbf{x}} = \underbrace{\begin{bmatrix} \cos(2\pi f t_1) & \sin(2\pi f t_1) \\ \vdots & \vdots \\ \cos(2\pi f t_N) & \sin(2\pi f t_N) \end{bmatrix}}_{\mathbf{H}} \underbrace{\begin{bmatrix} \alpha_1 \\ \alpha_2 \end{bmatrix}}_{\boldsymbol{\theta}} + \boldsymbol{\epsilon}, \tag{7}$$

where $\boldsymbol{\epsilon} \sim \mathcal{N}(\mathbf{0}, \sigma^2 \mathbf{I}_N)$. The maximum likelihood estimator of $\boldsymbol{\theta}$ is equivalent to the least-squares solution and is given by:

$$\widehat{\boldsymbol{\theta}} = (\mathbf{H}^\mathsf{T}\mathbf{H})^{-1}\mathbf{H}^\mathsf{T}\mathbf{x} \tag{8}$$

The above equation yields a solution for the estimators $\widehat{\alpha}_1$ and $\widehat{\alpha}_2$, which can readily be converted into the original parameters $A$ and $\phi$ as:

$$\widehat{A} = \sqrt{\frac{\left(\sum_{n=1}^N x_{t_n}\cos(2\pi f(t_n - \phi))\right)^2}{\sum_{n=1}^N \cos^2(2\pi f(t_n - \phi))} + \frac{\left(\sum_{n=1}^N x_{t_n}\sin(2\pi f(t_n - \phi))\right)^2}{\sum_{n=1}^N \sin^2(2\pi f(t_n - \phi))}} \tag{9}$$

$$\phi = \frac{1}{4\pi f}\tan^{-1}\left(\frac{\sum_{n=1}^N \sin(4\pi f t_n)}{\sum_{n=1}^N \cos(4\pi f t_n)}\right) \tag{10}$$

Note here that $\phi$ is not random, since it does not depend on the observed signal. Since we are working with a sinusoid, the amplitude estimator $\widehat{A}$ can be converted to a measure of power as $P(f) \propto \frac{\widehat{A}^2}{2}$ to provide the relative strength of the frequency $f$:

$$P(f) = \frac{1}{2}\left(\frac{\left(\sum_{n=1}^N x_{t_n}\cos(2\pi f(t_n - \phi))\right)^2}{\sum_{n=1}^N \cos^2(2\pi f(t_n - \phi))} + \frac{\left(\sum_{n=1}^N x_{t_n}\sin(2\pi f(t_n - \phi))\right)^2}{\sum_{n=1}^N \sin^2(2\pi f(t_n - \phi))}\right) \tag{11}$$

Repeating this exercise for a collection of candidate frequencies $f_1, \ldots, f_K$ (and yielding corresponding power estimators $P(f_1), \ldots, P(f_K)$), we can get obtain a periodogram of the signal. It can be shown that under Gaussian noise ($\sigma^2 = 1$), $P(f)$ follows a $\chi^2$ distribution with 2 degrees of freedom, allowing for ease of access to the false alarm probabilities of the power for each frequency.

## B. Theoretical Discussions

In this section, we provide a high-level theoretical discussion of our work.

## B.1. Conditional Entropy of Reverse Process

First, we leverage a result that was presented in (Yang et al., 2024a) related to the conditional entropy of the reverse process of a diffusion model. Specifically, the theoretical result presented by the authors shows that the conditional entropy of the diffusion reverse process given the frequency representation of time series is strictly less than the conditional entropy of the reverse diffusion process without the frequency information:

$$\mathbb{H}(\mathbf{X}_{t-1}^{ta}|\mathbf{X}_t^{ta}, \mathbf{X}_0^{co}, \mathbf{C^H}, \mathbf{C^D}) < \mathbb{H}(\mathbf{X}_{t-1}^{ta}|\mathbf{X}_t^{ta}, \mathbf{X}_0^{co}) \tag{12}$$

where $\mathbf{X}_t^{ta}$ denotes the diffusion process of the target at time $t$, $\mathbf{X}_0^{co}$ denotes the time-series observation condition, and $\mathbf{C^H}$ and $\mathbf{C^D}$ denote the high-frequency and dominant-frequency condition, respectively. Here, $\mathbb{H}(\cdot|\cdot)$ denotes the conditional entropy, which for two random quantities $\mathbf{X}$ given $\mathbf{Y}$ is defined as:

$$\mathbb{H}(\mathbf{X}|\mathbf{Y}) = -\int p(\mathbf{x}, \mathbf{y}) \log p(\mathbf{x}|\mathbf{y}) d\mathbf{x}, \tag{13}$$

where $p(\mathbf{x}, \mathbf{y})$ denotes the joint distribution of $\mathbf{X}$ and $\mathbf{Y}$, while $p(\mathbf{x}|\mathbf{y})$ denotes the conditional distribution of $\mathbf{X}$ given $\mathbf{Y}$. The essence behind the theoretical result shown in (12) is that incorporating additional conditional information reduces the entropy of the reverse process. This result also applies to conditioning on additional information, such as the encoded Lomb-Scargle representation $\mathbf{Z}_S = \mathcal{E}_{\text{spec}}(\mathcal{LS}(\mathbf{X}_0^{co}))$ and thus we can directly deduce that:

$$\mathbb{H}(\mathbf{X}_{t-1}^{ta}|\mathbf{X}_t^{ta}, \mathbf{X}_0^{co}, \mathbf{Z}_S) < \mathbb{H}(\mathbf{X}_{t-1}^{ta}|\mathbf{X}_t^{ta}, \mathbf{X}_0^{co}) \tag{14}$$

The proof of this result is straightforward and follows the same logic as the proof presented in Appendix A of (Yang et al., 2024a).

## B.2. Potential Bias of Sinusoidal Assumption of Lomb-Scargle

Recall from Appendix A, each frequency component of the Lomb-Scargle periodogram can be derived using least-squares, where the underlying time-series signal is assumed to be a sinusoid with zero-mean and additive noise. From the maximum likelihood point of view, the noise is Gaussian with variance $\sigma^2$. One important aspect of this work to discuss is the potential bias of utilizing the Lomb-Scargle periodogram in the imputation model, since the periodogram is based on the assumption that the underlying data is sinusoidal. In this work, the Lomb-Scargle periodogram is incorporated in two aspects: (1) as a conditioning variable in the imputation model; and (2) as part of the spectral consistency loss $\mathcal{L}_{\text{SCons}}$ used to fine-tune our model in the final stage of training to enforce alignment between the spectrum of the observed data and that of the generative model.

In the first aspect, the Lomb-Scargle periodogram is encoded using a transformer module and incorporated as a conditioning vector throughout the reverse process of the diffusion model. This conditioning provides spectral information about the target time series, guiding the generation process toward samples with similar frequency characteristics. Unlike conditioning on independent variables, the periodogram contains meaningful information about the spectral properties of the time series being generated, making it an informative conditioning signal that should improve generation quality. The sinusoid assumption of the time-series utilized in the derivation of the Lomb-Scargle periodogram does not introduce bias into the model, but rather guides the diffusion process. In the scenario in which the periodogram is not useful, the model would learn to ignore the conditional information. In our work, we have observed that utilizing this information does improve the generative quality.

In the second aspect, the spectral fine-tuning introduces a regularization of our model from the score-matching optimum obtained in the first stage of training, creating a trade-off between distributional accuracy (captured by the initial training) and spectral fidelity. To be precise, in the first part of training, the standard score-matching objective is optimized, that is,

$$\mathcal{L}(\theta) = \mathbb{E}\left[\|\boldsymbol{\epsilon} - \boldsymbol{\epsilon}_\theta(\mathbf{x}_t^{ta}, t \mid \mathbf{x}_0^{co})\|^2\right], \tag{15}$$

and in the second part of training, the spectral consistency loss is minimized, where:

$$\mathcal{L}_{\text{SCons}}(\theta) = \mathbb{E}\left[\|\mathcal{LS}(\mathbf{x}_0^{co}) - \mathcal{LS}(\widehat{\mathbf{x}}_0^{co})\|_2^2\right], \tag{16}$$

While the score-matching objective learns to approximate the full data distribution, it may not adequately capture the frequency representation of the time series data. The spectral consistency loss addresses this limitation by explicitly

enforcing that generated samples maintain spectral characteristics consistent with the observed data, which we believe is especially useful for time-series with high degrees of missingness. One can roughly interpret our training algorithm as minimizing a single regularized loss with two terms:

$$\mathcal{L}_{\text{reg}}(\theta) = \lambda_1 \mathcal{L}(\theta) + \lambda_2 \mathcal{L}_{\text{SCons}}(\theta), \tag{17}$$

where $\lambda_1 > 0$ and $\lambda_2 > 0$ are scaling constants. As can be seen in (17), our training algorithm will introduce a bias, since the spectral consistency loss will deviate the parameters from optimal solution, which minimizes the score-matching objective.

## C. Computation times

We present an analysis of the computational speed of our method (LSCD). In Tables 4 and 5 we show a comparison of training and inference computation times respectively, for the LSCD and CSDI models, evaluated on PhysioNet and PM2.5 datasets. All computations in this analysis were performed using a g5.2xlarge AWS instance (AMD EPYC 7R32 CPU, with an Nvidia A10G 24 GB GPU). As shown in the tables, the percentage increase in computation time is approx. 9% for training and 13% for inference. However, the training of LSCD includes a final fine-tuning stage using the spectral consistency loss $\mathcal{L}_{SCons}$ from Section 4.3, which takes $288.7\,s/ep$ for PhysioNet and $430.3\,s/ep$ for PM2.5, due to requiring running the inference pipeline as part of the computation. Considering this step together with the full training process, LCSD resulted in an additional 43% training time for PhysioNet and 45% for PM2.5.

Table 4: Training time per epoch for CSDI and LSCD. The last column shows the relative increase for LSCD. Measurements were averaged over 10 epochs.

|  | CSDI (s/epoch) | LSCD (s/epoch) | $\Delta$Time (%) |
|---|---|---|---|
| PhysioNet | 10.30 | 11.18 | 8.5% |
| PM2.5 | 14.82 | 16.23 | 9.5% |

Table 5: Inference time per batch for CSDI and LSCD (batch size = 16). The last column shows the relative increase for LSCD. Measurements were averaged over 5 batches.

|  | CSDI (s/batch) | LSCD (s/batch) | $\Delta$Time (%) |
|---|---|---|---|
| PhysioNet | 88.19 | 99.18 | 13.3% |
| PM2.5 | 69.36 | 78.58 | 12.5% |

## D. Datasets

Here we introduce in detail the datasets used in our evaluation. We evaluate our method on both synthetic and real-world datasets with varying degrees of missingness to assess its robustness in imputation tasks. We use publicly available real data from two domains, healthcare and climate, and we generate a synthetic dataset where we have control over the ground truth frequencies. A summary of the datasets characteristics is shown in Table 6

Table 6: Summary of datasets used for evaluation.

| DATASET | # SAMPLES | # FEATURES | TIME STEPS | MISSINGNESS TYPE | MISSING % | LINK |
|---|---|---|---|---|---|---|
| SYNTHETIC SINES | 2000 | 5 | 100 | MCAR, SEQUENCE, BLOCK | 10% | N/A |
| PHYSIONET | 4000 | 35 | 48 | NATURALLY OCCURRING + MCAR | 80% (NATURAL) | [LINK] |
| PM2.5 AIR QUALITY | 5633 | 36 | 36 | NON-RANDOM + ARTIFICIAL | 13% (NATURAL) | [LINK] |

**PhysioNet.** This dataset contains multivariate physiological measurements from ICU patients (Silva et al., 2012). We use the preprocessed version from Cao et al. (2018); Tashiro et al. (2021), consisting of 4,000 patient records with 35 variables measured hourly over 48 time steps. Approximately 80% of values are naturally missing. For evaluation, we hold out $10\%$, $50\%$, and $90\%$ of the observed values as ground truth and assess imputation quality on these missing entries.

**PM2.5 Air Quality Data.** This dataset contains hourly PM2.5 pollution measurements from 36 monitoring stations in Beijing over a 12-month period (Yi et al., 2016). Following prior work (Cao et al., 2018; Tashiro et al., 2021), we extract sequences of length 36. The dataset exhibits approximately 13% naturally occurring missing values, with additional structured missingness artificially introduced for evaluation. The dataset presents real-world challenges such as non-random missing patterns and periodic fluctuations in pollution levels.

### D.1. Synthetic Sines dataset

We generate a synthetic time series dataset for evaluating the proposed method. The dataset is designed to simulate multivariate time series with distinct frequency and amplitude characteristics across different channels. The process is described below.

#### D.1.1. TIME SERIES GENERATION

Let $L$ represent the number of timesteps, $K$ the number of channels, and $T$ the maximum time horizon. For each sample, a multivariate time series $X \in \mathbb{R}^{L \times K}$ is generated using a sum of sinusoidal components with additive Gaussian noise. Specifically, the time series for each channel $k$ is generated as follows:

$$f_k(t) = \sum_{i=1}^{N_k} a_{k,i} \sin(2\pi f_{k,i} t + \phi_{k,i}) + \varepsilon_k,$$

where $a_{k,i}$, $f_{k,i}$, and $\phi_{k,i}$ are the amplitude, frequency, and phase of the $i$-th sinusoidal component of channel $k$, respectively. The term $\varepsilon_k$ represents Gaussian noise with zero mean and a standard deviation $\sigma_k$. The number of sinusoidal components $N_k$, as well as the parameters $a_{k,i}$, $f_{k,i}$, and $\phi_{k,i}$, vary across channels to simulate heterogeneity.

#### D.1.2. FREQUENCY AND AMPLITUDE SAMPLING

For each channel $k$, the amplitudes $a_{k,i}$ are fixed for all samples within the dataset, while the frequencies $f_{k,i}$ are drawn from a Beta distribution, defined as:

$$f_{k,i} \sim \text{Beta}(\alpha = 2, \beta = 2) \cdot w_k + \left(\mu_k - \frac{w_k}{2}\right),$$

where $\mu_k$ and $w_k$ represent the mean and width of the frequency range for channel $k$. Sampling from a Beta distribution ensures that frequencies are concentrated around $\mu_k$ but still allow variability, making the generated time series more realistic. This approach follows Fourier Flows (Alaa et al., 2021), which leverages Beta distributions for frequency modeling in generative time series tasks. Fixing the amplitudes $a_{k,i}$ for each frequency simplifies the interpretation of each sinusoidal component's contribution and reduces the complexity of parameter selection while retaining diversity in frequency combinations.

#### D.1.3. DATASET CHARACTERISTICS

The dataset contains 2000 samples, each with $L = 100$ timesteps spanning a time horizon of $T = 10.0$ units. The dataset consists of $K = 5$ channels with distinct amplitude and frequency configurations, summarized in Table 7.

Table 7: Characteristics of the generated dataset for each channel.

| Channel | Components ($N_k$) | Mean Frequencies ($\mu_k$) | Frequency Widths ($w_k$) | Amplitudes ($a_{k,i}$) |
|---------|--------------------|----------------------------|--------------------------|------------------------|
| 1 | 1 | [1.0] | [1.0] | [1.0] |
| 2 | 2 | [1.0, 2.0] | [1.0, 1.5] | [0.5, 1.0] |
| 3 | 3 | [1.0, 2.0, 3.0] | [1.0, 1.0, 1.5] | [0.5, 1.0, 1.5] |
| 4 | 4 | [0.5, 1.0, 1.5, 2.0] | [1.0, 1.0, 1.0, 2.0] | [0.8, 1.2, 1.5, 2.0] |
| 5 | 5 | [0.5, 1.0, 2.0, 3.0, 4.0] | [0.5, 1.0, 1.0, 1.5, 2.0] | [1.0, 1.5, 2.0, 2.5, 3.0] |

#### D.1.4. MISSINGNESS SIMULATION

A binary mask $M \in \{0,1\}^{L \times K}$ is generated to simulate missingness in the dataset. To make the setting more realistic, the dataset is initially generated with 10% missing data, where values are randomly masked. This ensures that even before

applying specific missingness mechanisms, the dataset already contains some level of real-world uncertainty.

We evaluate three missingness mechanisms:

- **MCAR (Missing Completely at Random)**: Values are masked independently of the underlying data distribution. This serves as a baseline missingness pattern.

- **Sequence Missing**: Contiguous time intervals are masked to simulate sensor downtimes or transmission failures.

- **Block Missing**: Large regions of the dataset, spanning both time steps and feature dimensions, are masked to mimic large-scale data outages. The actual missing rate of block missingness is difficult to strictly control due to the overlap between blocks. Instead, a "factor" is used to adjust the missing rate approximately.

Recent studies on time series imputation (Du et al., 2024; Mitra et al., 2023) highlight that block and sequence missingness patterns better represent real-world scenarios compared to MAR or MNAR mechanisms. These structured patterns closely align with real applications such as sensor failures, data transmission losses, and large-scale system outages. Unlike MCAR, which assumes uniform randomness, structured missingness provides a more challenging and realistic evaluation of imputation models. Furthermore, methods that perform well under sequence and block missingness conditions tend to generalize better across diverse real-world datasets.

We use the `PyGrinder`[1] library to generate missingness patterns in our dataset, ensuring a standardized and reproducible approach to missing data simulation.

Each missingness mechanism is applied at three levels (10%, 50%, and 90%) to assess the robustness of our method under increasing data loss. The specific hyperparameters used for each mechanism and the resulting missing rates, as computed from the actual data, are summarized in Table 8.

Table 8: Missingness parameters and actual missing rates for different mechanisms.

| Mechanism | Hyperparameters | Target Rate | Actual Rate |
|-----------|-----------------|-------------|-------------|
| MCAR | $p = 0.1$ | 10% | 10.0% |
| MCAR | $p = 0.5$ | 50% | 50.0% |
| MCAR | $p = 0.9$ | 90% | 90.0% |
| Sequence Missing | $p = 0.1$, seq_len $= 50$ | 10% | 10.0% |
| Sequence Missing | $p = 0.5$, seq_len $= 50$ | 50% | 50.0% |
| Sequence Missing | $p = 0.9$, seq_len $= 30$ | 90% | 55.6% |
| Block Missing | factor $= 0.1$, block_len $= 40$, block_width $= 4$ | 10% | 18.8% |
| Block Missing | factor $= 0.5$, block_len $= 40$, block_width $= 4$ | 50% | 56.4% |
| Block Missing | factor $= 0.9$, block_len $= 40$, block_width $= 4$ | 90% | 71.9% |

As seen in Table 8, the actual missing rates for block missingness tend to deviate from the target values. This is due to the nature of block-based masking, where overlapping blocks and feature constraints introduce variability in the effective missing proportion. Instead of directly specifying the missing rate, the "factor" parameter is used to approximate the final missing rate, though it does not always match the intended level exactly. The structured missingness patterns allow for evaluating the model's ability to reconstruct both small-scale gaps and large missing regions effectively.

---

[1] https://github.com/WenjieDu/PyGrinder

# E. Implementation and Reproducibility details

## E.1. Lomb-Scargle implementation

The following listing provides a PyTorch implementation of the Lomb–Scargle periodogram, designed to efficiently compute spectral estimates for irregularly sampled time series. This implementation is fully differentiable, allowing seamless integration into learning-based models for gradient-based optimization.

```python
import torch

class LombScargleBatchMask(torch.nn.Module):
    def __init__(self, omegas):
        super(LombScargleBatchMask, self).__init__()
        self.omegas = omegas  # Tensor of angular frequencies (omegas)

    def forward(self, t, y, mask=None):
        """
        Computes the Lomb-Scargle periodogram for a batch of time series with masking.

        Args:
            t (Tensor): Time values, shape [B, N].
            y (Tensor): Observed data values, shape [B, N].
            mask (Tensor, optional): Boolean mask, shape [B, N] (1 = valid, 0 = missing).

        Returns:
            Tensor: Lomb-Scargle periodogram values, shape [B, M].
        """
        B, N = t.shape
        M = self.omegas.shape[0]

        if mask is None:
            mask = torch.ones_like(t)  # Default to all valid points

        # Expand tensors
        t = t.unsqueeze(1)  # [B, 1, N]
        y = y.unsqueeze(1)  # [B, 1, N]
        mask = mask.unsqueeze(1)  # [B, 1, N]
        omega = self.omegas.view(1, M, 1)  # [1, M, 1]

        # Compute tau for each frequency and batch
        two_omega_t = 2 * omega * t  # [B, M, N]
        sin_2wt = (torch.sin(two_omega_t) * mask).sum(dim=2)
        cos_2wt = (torch.cos(two_omega_t) * mask).sum(dim=2)

        tan_2omega_tau = sin_2wt / torch.clamp(cos_2wt, min=1e-10)
        tau = torch.atan(tan_2omega_tau) / (2 * torch.clamp(omega.squeeze(2), min=1e-10))

        # Compute Lomb-Scargle periodogram
        omega_t_tau = omega * (t - tau.unsqueeze(2))  # [B, M, N]
        cos_omega_t_tau = torch.cos(omega_t_tau) * mask
        sin_omega_t_tau = torch.sin(omega_t_tau) * mask

        y_cos = y * cos_omega_t_tau
        y_sin = y * sin_omega_t_tau

        P_cos =(y_cos.sum(dim=2)**2)/torch.clamp(cos_omega_t_tau.pow(2).sum(dim=2),min=1e-10)
        P_sin = (y_sin.sum(dim=2)**2)/torch.clamp(sin_omega_t_tau.pow(2).sum(dim=2),min=1e-10)

        P = 0.5 * (P_cos + P_sin)  # [B, M]

        return P
```

Listing 1: Batch Lomb-Scargle Periodogram with Masking

## E.2. Model Architecture Details

**Diffusion Model Architecture**

Our approach adopts a *diffusion-based generative model* for time series imputation, implemented in a style consistent with standard score-based diffusion frameworks (Tashiro et al., 2021). Below, we outline the high-level architecture and key components:

- **Denoising Network:** A U-shaped transformer that incorporates temporal convolution blocks and multi-head attention layers. The network accepts the following inputs:

1. Noisy target values $\mathbf{x}_t^{ta}$ at diffusion step $t$,
2. Observed context $\mathbf{x}_0^{co}$ (unmasked entries),
3. Spectral embeddings $\mathbf{z}_S$ from the Lomb–Scargle encoder,
4. A positional encoding of the diffusion step $t$.

- **Irregular Sampling Handling:** Timestamps $\mathbf{s}$ are mapped through 128-dimensional positional embeddings and fused into the feature representations at each layer, enabling the model to account for non-uniform time intervals.

- **Network Block Details:** For each denoising step, the architecture employs a series of *transformer blocks* to capture temporal structure:

  - *Multi-Head Self-Attention*, which learns dependencies among time steps,
  - *Positional Encodings* added along the time axis to preserve sequence order,
  - *Feed-Forward* layers with dropout and normalization to encourage stable training and robust generalization.

### Spectrum Encoder Transformer Blocks

As described in Section 4.2, we employ two transformer modules, one along the frequency axis, and one along the features axis, in order to extract a latent embedding $\mathbf{z}_S$ from the Lomb–Scargle periodogram $\mathcal{LS}(\mathbf{x}_0^{co}) \in \mathbb{R}^{K \times L}$. Key hyperparameters include:

- $\mathbf{d}_{\mathrm{model}}$ (internal hidden dimension used by each attention module): $64$.

- $\mathrm{nHeads}$ (number of attention heads): $8$.

- $\mathrm{depth}$ (number of transformer layers): $4$ blocks for each dimension (frequency/features).

**Input Representation:** Log-transformed power spectral density values $\log(1 + \mathcal{LS}(\mathbf{x}_0^{co}))$ across $J$ frequency bins, normalized to zero mean/unit variance per channel.

### Training and Hyperparameters

We train for 400 epochs and select the best checkpoint via a validation set. The final $\mathbf{z}_S$ from our spectral encoder is concatenated with the other conditioning signals (e.g. partial observations) at *every* denoising step to guide the model.

- **Diffusion Steps**: $T_{\max} = 50$.

- **Batch Size**: 16.

- **Learning Rate**: 1e-3.

## F. Evaluation Metrics

To ensure a fair evaluation, we compute reconstruction errors only on the target values, defined as the set of originally observed values that were artificially removed for evaluation. In contrast, frequency-domain metrics are computed using all observed values (condition plus target), ensuring that spectral estimates are not influenced by imputed values outside of the ground truth.

In line with the conditional diffusion framework described in Section 3.3, let us recall that we split the data into a *conditional* portion $\boldsymbol{x}_0^{co} = \boldsymbol{m}^{co} \odot \boldsymbol{X}$ and a *target* portion $\boldsymbol{x}_0^{ta} = (\boldsymbol{M} - \boldsymbol{m}^{co}) \odot \boldsymbol{X}$. When we refer to "originally observed values that were artificially removed," we mean $\boldsymbol{x}_0^{ta}$. At test time, the model imputes these missing target entries, producing a complete reconstruction $\hat{\boldsymbol{X}}$.

### F.1. Time-Domain Metrics

For time-domain metrics, we evaluate performance *only* on the target (imputed) entries, i.e., the set of indices where $(\boldsymbol{M} - \boldsymbol{m}^{co})$ is 1.

**Mean Absolute Error (MAE)**

$$\text{MAE} = \frac{\sum_{k,l}\left[\left(M_{k,l} - m_{k,l}^{co}\right)\left|X_{k,l} - \hat{X}_{k,l}\right|\right]}{\sum_{k,l}\left(M_{k,l} - m_{k,l}^{co}\right)}. \tag{18}$$

MAE measures the average absolute deviation between the imputed values $\hat{X}_{k,l}$ and the ground truth $X_{k,l}$ where $(M_{k,l} - m_{k,l}^{co}) = 1$.

**Root Mean Squared Error (RMSE)**

$$\text{RMSE} = \sqrt{\frac{\sum_{k,l}\left[\left(M_{k,l} - m_{k,l}^{co}\right)\left(X_{k,l} - \hat{X}_{k,l}\right)^2\right]}{\sum_{k,l}\left(M_{k,l} - m_{k,l}^{co}\right)}}. \tag{19}$$

RMSE penalizes large deviations more severely than MAE, again computed only over the target entries.

### F.2. Frequency-Domain Metrics

For frequency-domain metrics, we compute the Lomb–Scargle power spectral density (PSD) using only the *originally observed points*, i.e., where $M_{k,l} = 1$. This ensures the spectral estimate is not influenced by imputed values outside the known ground truth.

**Spectral Mean Absolute Error (S-MAE)**  To assess how well the imputed signals preserve the original frequency characteristics, we compare the normalized Lomb–Scargle PSD estimates of the ground-truth series $P_{\text{GT}}(\omega)$ against those of the reconstruction $P_{\text{Pred}}(\omega)$. Let $\Omega$ be the set of evaluated frequencies:

$$\text{S-MAE} = \frac{1}{|\Omega|} \sum_{\omega \in \Omega} \left| \frac{P_{\text{GT}}(\omega)}{\sum_{\omega'} P_{\text{GT}}(\omega')} - \frac{P_{\text{Pred}}(\omega)}{\sum_{\omega'} P_{\text{Pred}}(\omega')} \right|. \tag{20}$$

Here, $P_{\text{GT}}(\omega)$ and $P_{\text{Pred}}(\omega)$ are both computed strictly from points where $M_{k,l} = 1$.

# G. Visualizations of spectral distributions

## G.1. PM25 dataset

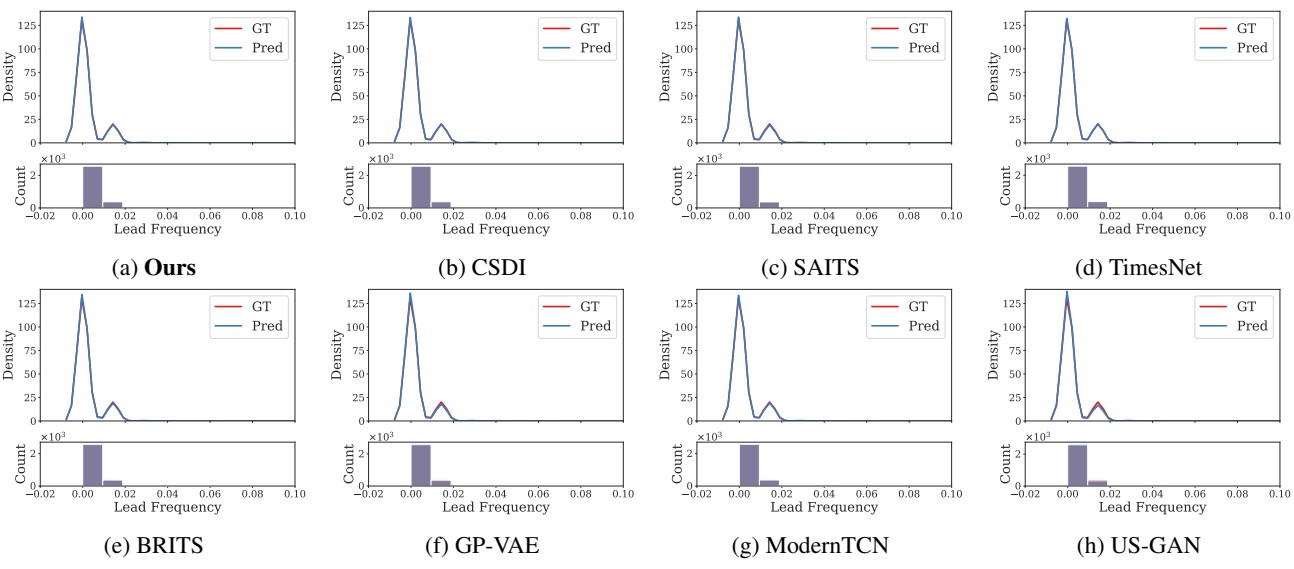

Figure 6: Distribution of leading frequency on PM25 dataset.

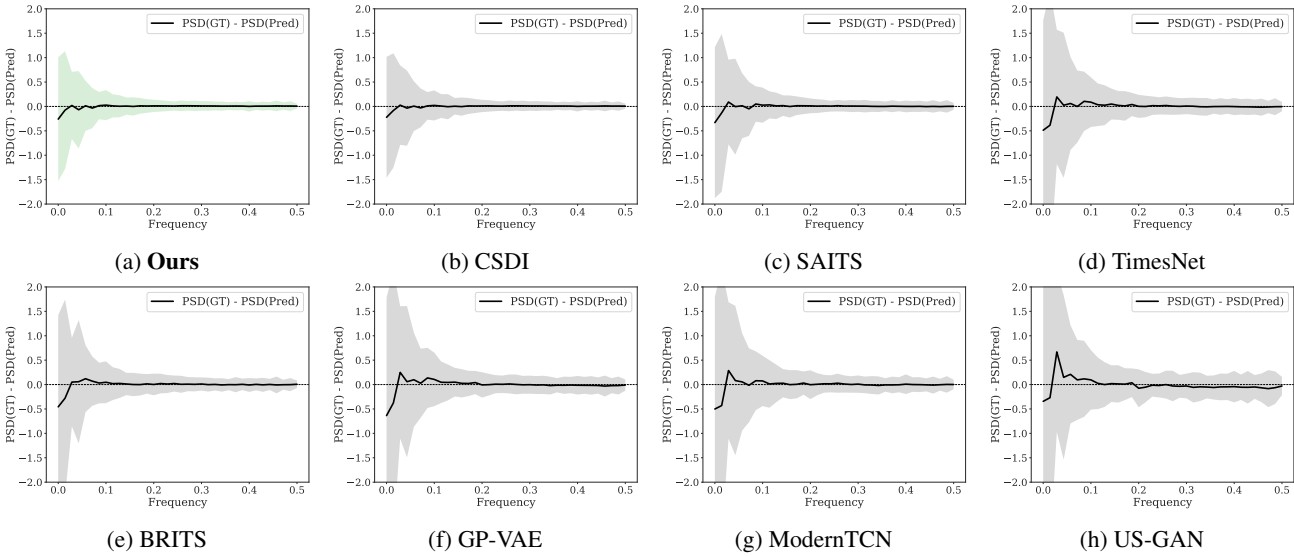

Figure 7: Difference between the power spectral densities (PSD) of the ground truth and predictions, with the black line representing the mean difference and the shaded area indicating one standard deviation

## G.2. Physionet dataset

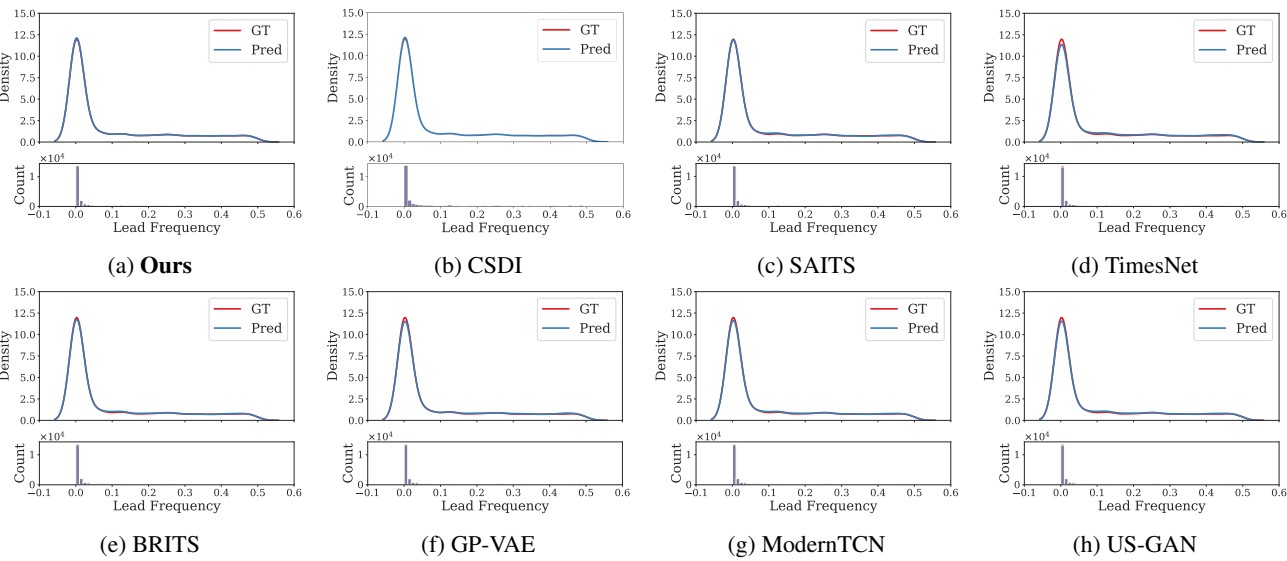

Figure 8: Distribution of leading frequency on Physio dataset with 10% missing data.

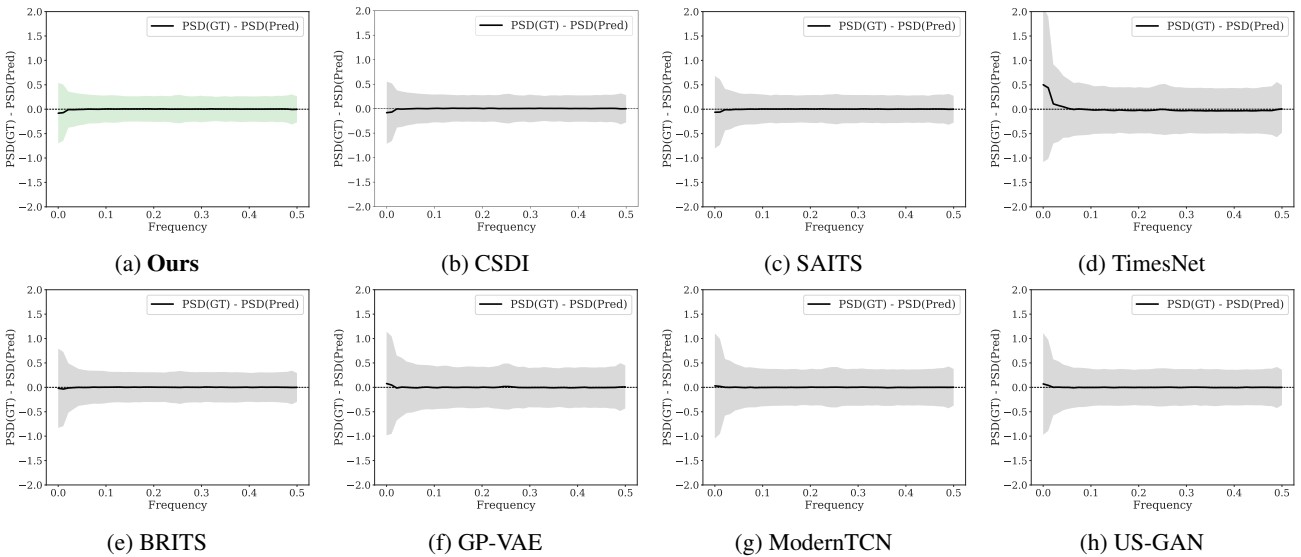

Figure 9: Difference between the power spectral densities (PSD) of the ground truth and predictions, with the black line representing the mean difference and the shaded area indicating one standard deviation, on Physionet with 10% missing data.

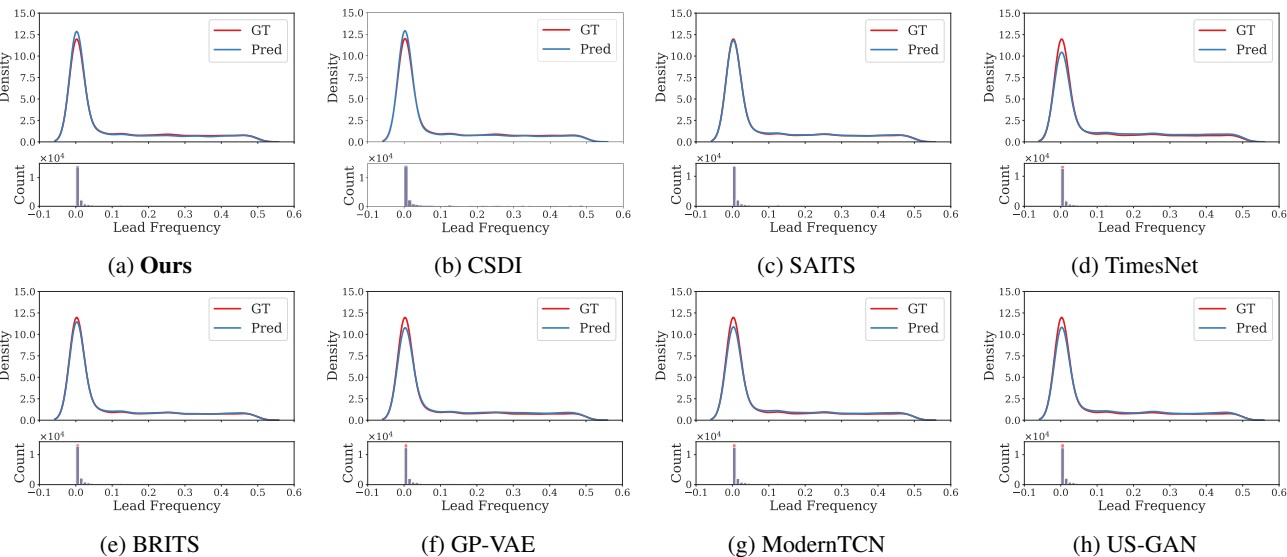

Figure 10: Distribution of leading frequency on Physio dataset with 50% missing data.

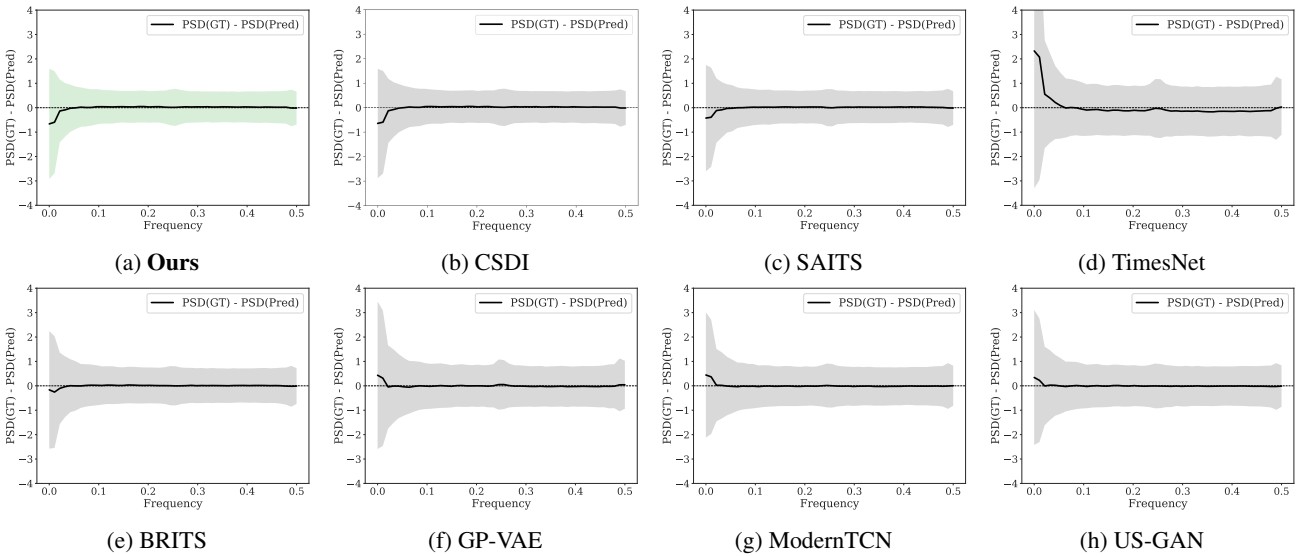

Figure 11: Difference between the power spectral densities (PSD) of the ground truth and predictions, with the black line representing the mean difference and the shaded area indicating one standard deviation, on Physionet with 50% missing data.

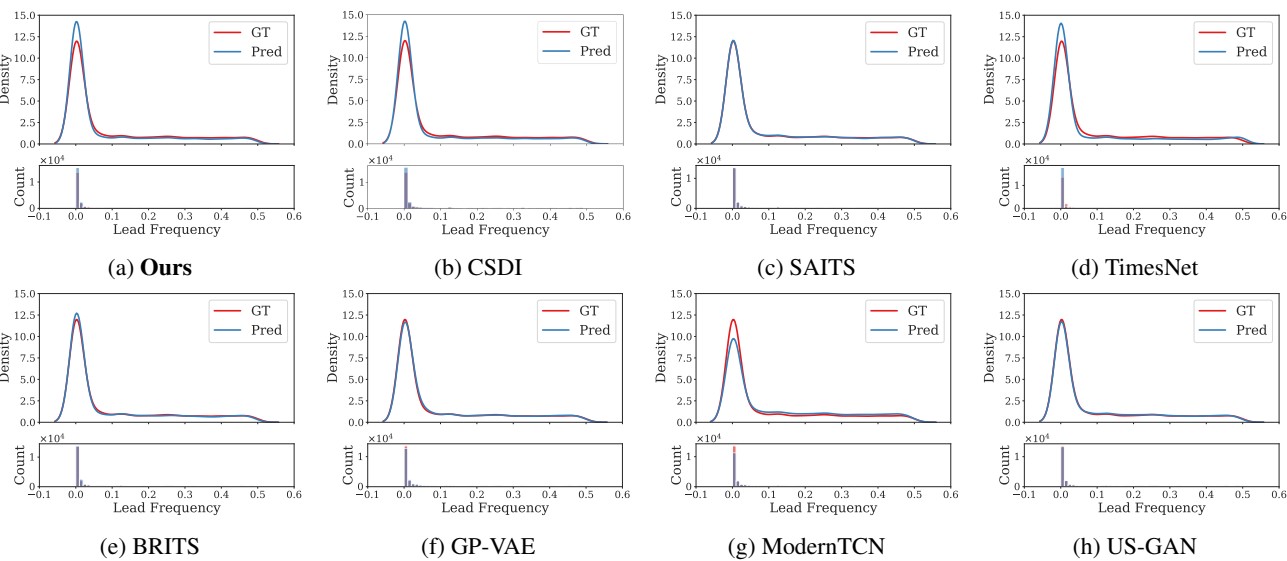

Figure 12: Distribution of leading frequency on Physio dataset with 90% missing data.

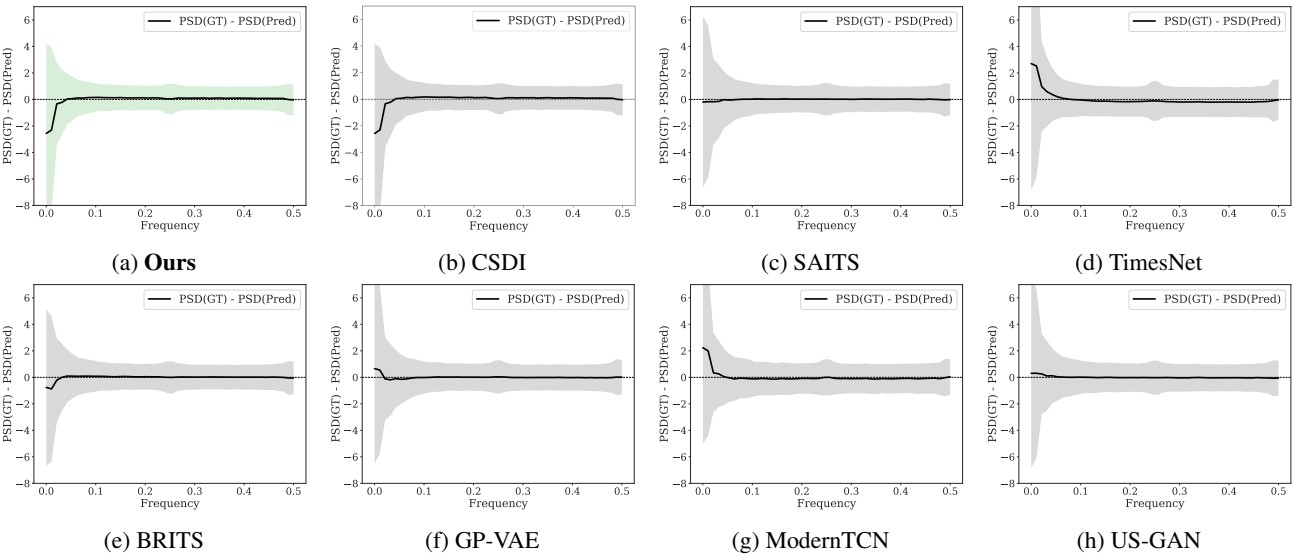

Figure 13: Difference between the power spectral densities (PSD) of the ground truth and predictions, with the black line representing the mean difference and the shaded area indicating one standard deviation, on Physionet with 90% missing data.

# H. Visualization of time distributions

## H.1. PM25 dataset

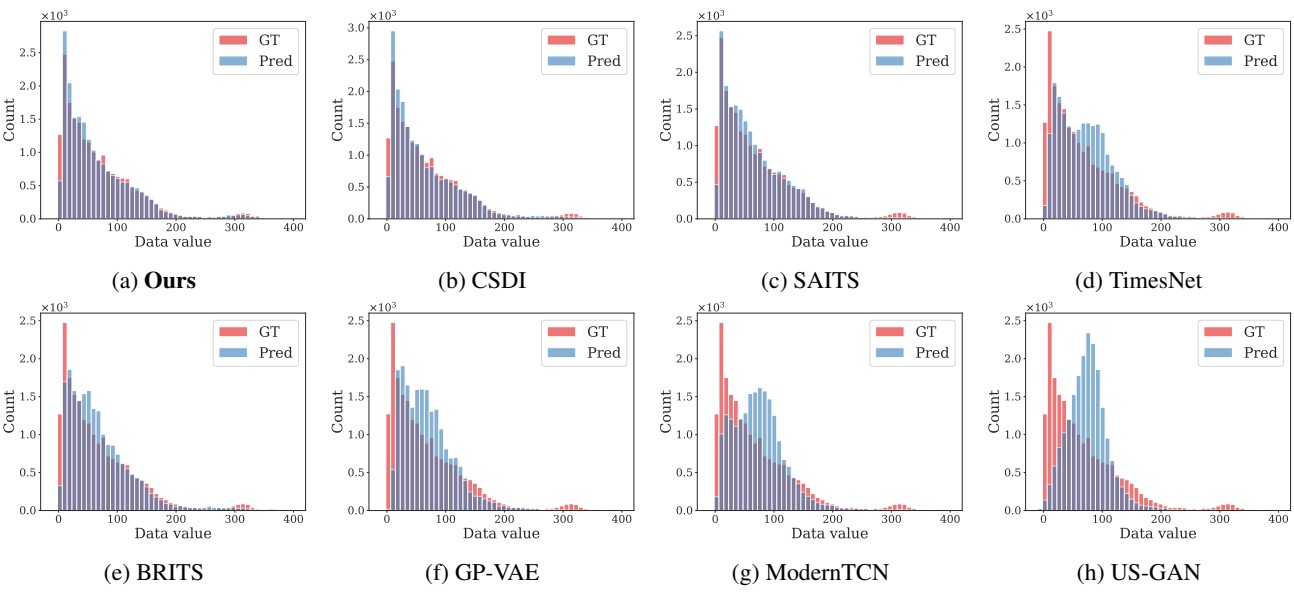

Figure 14: Distribution of imputed values across different models on PM25 dataset.

## H.2. Physionet dataset

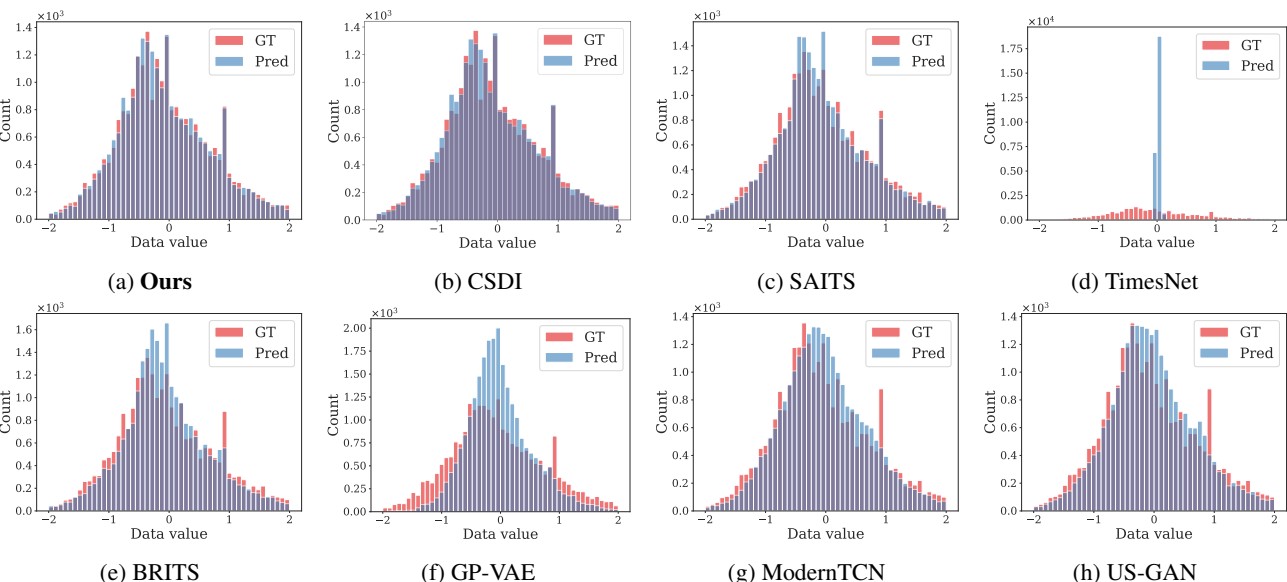

Figure 15: Distribution of imputed values across different models on on Physionet with 10% missing data.

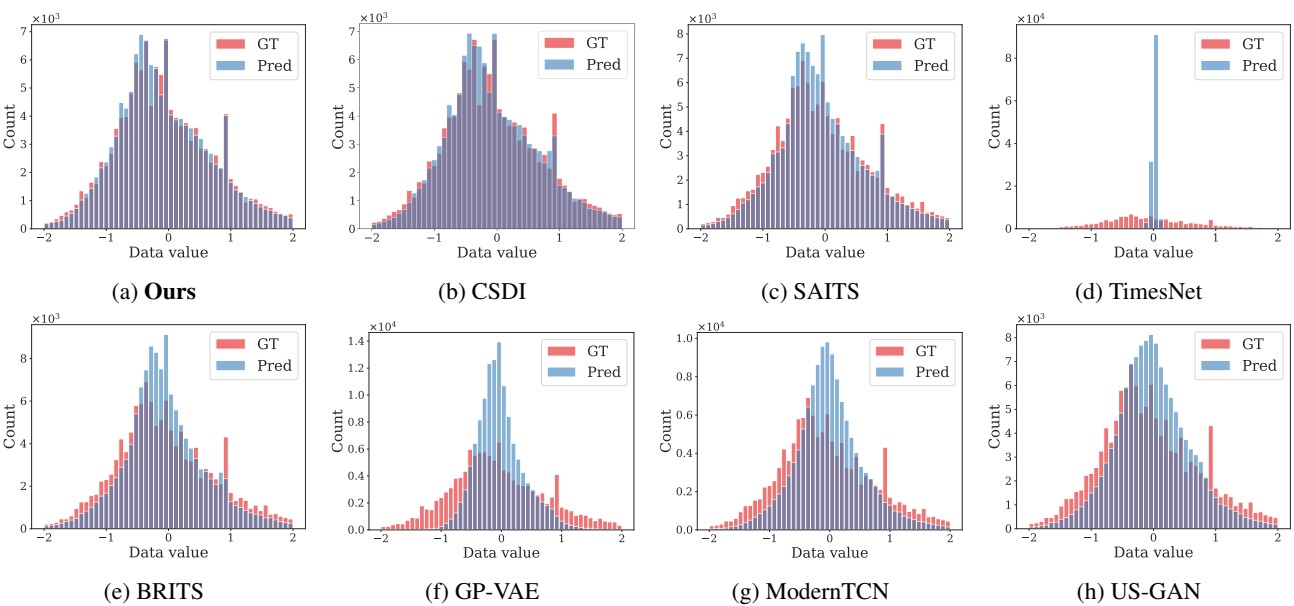

Figure 16: Distribution of imputed values across different models on on Physionet with 50% missing data.

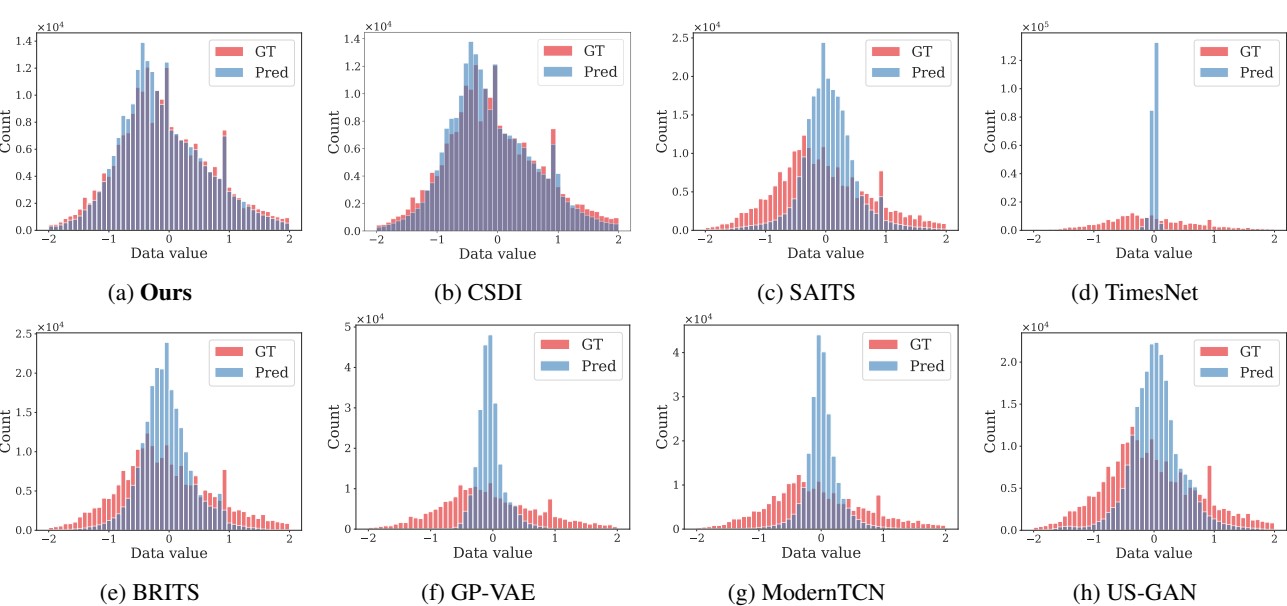

Figure 17: Distribution of imputed values across different models on on Physionet with 90% missing data.

# I. Visualization of imputed examples

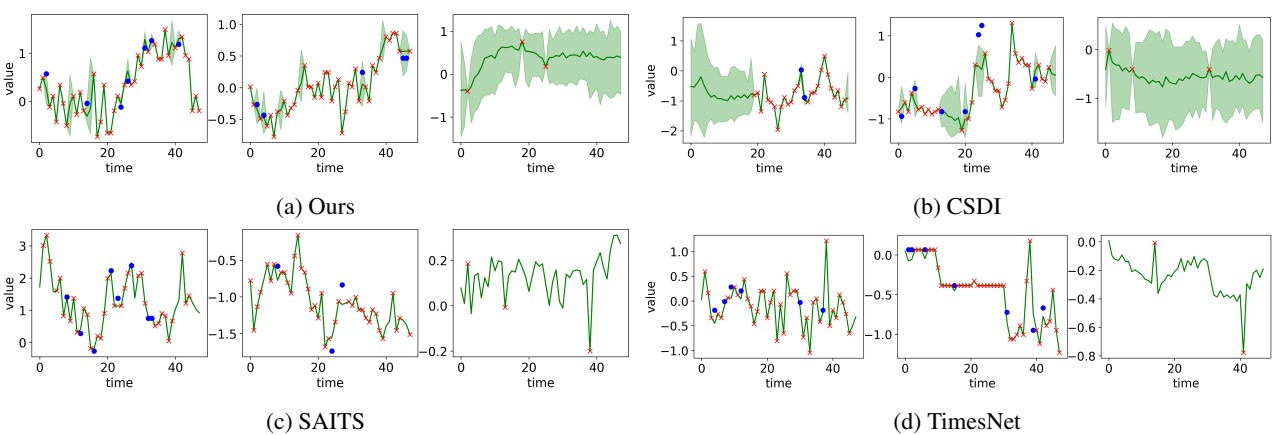

Figure 18: Reconstruction results on PhysioNet with 10% missing data.

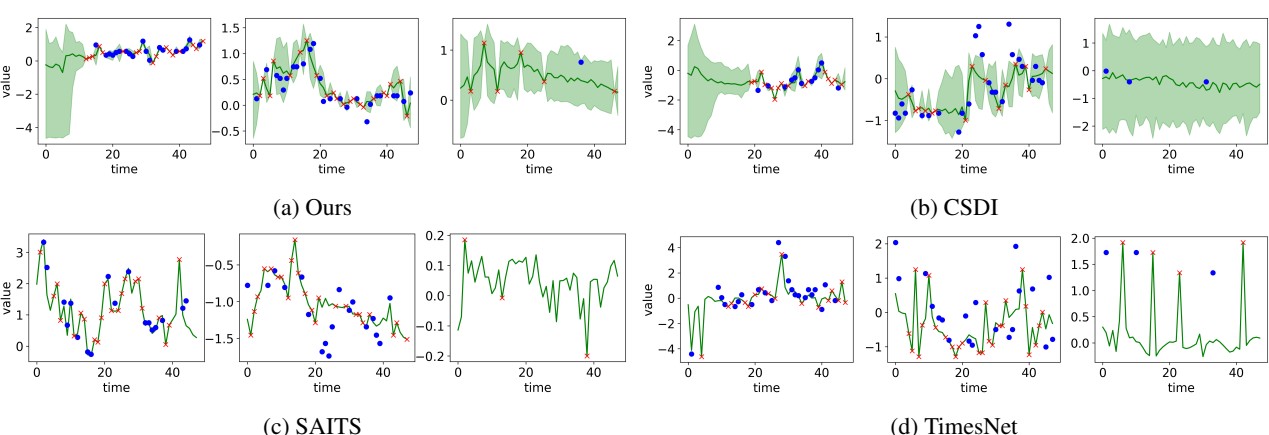

Figure 19: Reconstruction results on PhysioNet with 50% missing data.

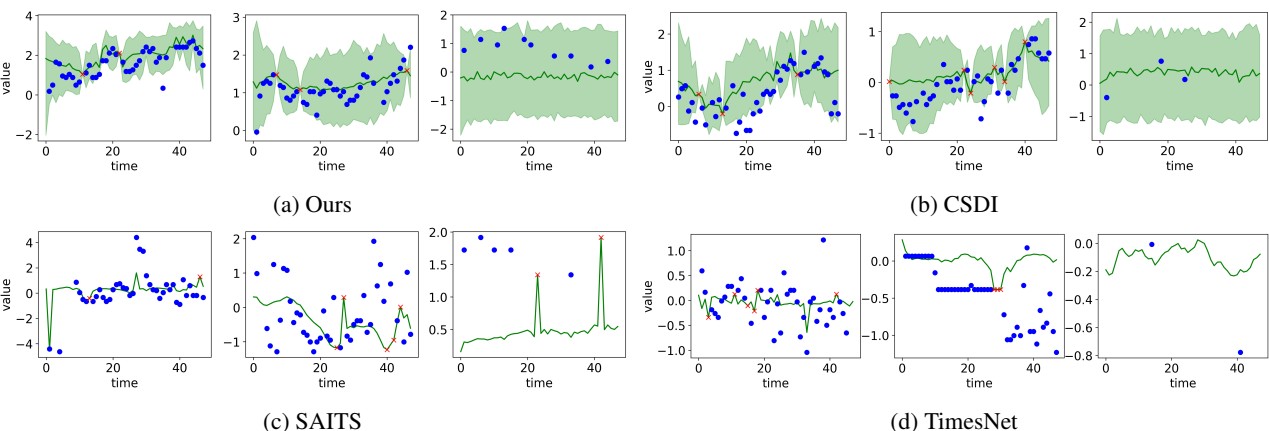

Figure 20: Reconstruction results on PhysioNet with 90% missing data.

