# OpenReview forum: "LSCD: Lomb--Scargle Conditioned Diffusion for Time series Imputation"
_ICML.cc/2025/Conference — ICML 2025 poster_

### Official Review · Reviewer_y6X7 · 2025-03-07

**Overall Recommendation:** 2

**Summary:**

The paper proposes LSCD for irregular time series imputation, which uses Lomb-Scargle periodograms to compute additional frequency domain loss in the diffusion model. LSCD uses Lomb-Scargle instead of FFT to transform irregular time series into the frequency domain and achieves more accurate imputation results.

**Claims And Evidence:**

From line 21 to line 23, the abstract claims that the method imputes without requiring imputation "in the frequency domain", which seems to conflict with the statement "prior to frequency estimation" from line 17 to line 18. Also, imputing in the time domain instead of the frequency domain is more prevalent, so I suspect the claim is wrong.

**Essential References Not Discussed:**

N/A

**Experimental Designs Or Analyses:**

Experimental designs are not convincing.
First, Lomb-Scargle is an approach relying heavily on the assumption of sinusoidal signal, which makes it unfair to compare the baselines on synthetic sines datasets. Also, unlike fully observed regular datasets, which usually exhibit clear seasonal patterns, real-world irregular time series usually lack such patterns, making them significantly different from synthetic sines datasets. Moreover, I think frequency domain patterns are not directly associated with data distributions shown in Figure 5, where data distributions are just statistical results on observed values. Therefore, visualizing data distributions cannot prove the effectiveness of  Lomb-Scargle on real-world irregular time series datasets. Lastly, since Lomb-Scargle is more computationally expensive than FFT, an analysis of training time and inference time is necessary to evaluate the model.

**Methods And Evaluation Criteria:**

In Figure 2, there are two questionable designs. First, in the calculation of consistency loss, the ground-truth spectrum is calculated using masked "observed condition" instead of the original "observed time series". Second, "imputation target" and "noisy imputation target" are plotted mostly the same, which is confusing. Since the code is unavailable in the provided anonymous github link, and LSCD is based on CSDI [1], I assume LSCD uses a similar approach in handling the diffusion and denoising process. According to the implementation of CSDI, "noisy imputation target" should be distorted with random noise instead of shifted compared to the "imputation target".
[1] Y. Tashiro, J. Song, Y. Song, and S. Ermon, "CSDI: Conditional Score-based Diffusion Models for Probabilistic Time Series Imputation," in Advances in Neural Information Processing Systems, Curran Associates, Inc., 2021, pp. 24804–24816.

**Other Comments Or Suggestions:**

N/A

**Other Strengths And Weaknesses:**

Other Weaknesses:

• The code is not available in the provided URL at line 34.

• Math symbols in Figure 2 are in low resolution.

• I believe irregular time series are not equal to time series with missing values. The paper mainly discusses the missingness scenarios in Figure 3. However, unalignments and irregular time intervals in real-world irregular time series are caused by different sampling rates of different variables, which is not the same as regularly sampled time series with missing values. Therefore, Using incomplete synthetic data to simulate irregular time series is unreasonable. Although CSDI [1] also uses irregular time series datasets to prove its effectiveness, its task is "probabilistic time series imputation", which is a more general task that includes irregular time series imputation.

[1] Y. Tashiro, J. Song, Y. Song, and S. Ermon, "CSDI: Conditional Score-based Diffusion Models for Probabilistic Time Series Imputation," in Advances in Neural Information Processing Systems, Curran Associates, Inc., 2021, pp. 24804–24816.

**Questions For Authors:**

What's the research purpose of irregular time series imputation, especially on medical datasets? For example, in Physionet'12, samples are collected during patients' 48 hours of stay in ICU, where the values of different features are sampled at different frequencies as scheduled. The missingness is informative to indicate different sampling rates. Therefore, imputation for these medical irregular times series seems to lack proper motivations.

**Relation To Broader Scientific Literature:**

The main contribution is the usage of frequency domain information in the diffusion model. However, compared to previous work CSDI [1], the introduction of the diffusion model in Figures and the experimental comparisons are not convincing enough.

[1] Y. Tashiro, J. Song, Y. Song, and S. Ermon, "CSDI: Conditional Score-based Diffusion Models for Probabilistic Time Series Imputation," in Advances in Neural Information Processing Systems, Curran Associates, Inc., 2021, pp. 24804–24816.

**Theoretical Claims:**

I check the correctness of conditional diffusion in section 3.3.

---

> ### Author Rebuttal · Authors · 2025-04-01
>
> We thank the reviewer for their careful assessment of our paper. Below, we provide detailed responses aimed at clarifying and addressing each concern.
>
> &nbsp;
> ### Claims and Evidence
> Thank you for pointing out the discrepancy: in line 22, "frequency domain" should be "time domain". We have updated the text accordingly.
>
> &nbsp;
> ### Methods And Evaluation Criteria (Figure 2)
> **Consistency loss:** We can confirm that the diagram correctly reflects the architecture, and that using $x_{co}$ instead of the "observed time series" is done by design: during training, we sample a conditional mask $m_{co}$ from the observed values and treat them as $x_{co}$. The consistency loss then ensures the reconstruction matches the frequency profile of these masked ‘observed’ values, which simulates the inference setting. This follows common practice in score-based diffusion training, including CSDI.
>
> **Noisy imputation target:** We have updated the figure so the effect of adding noise is more evident and does not resemble a shift.
>
> **Math symbols:** We have now increased the resolution of math symbols in the figure.
>
> &nbsp;
> ### Experimental Design or Analyses
> **Use of Sine Datasets:** We acknowledge that the Lomb-Scargle method is well-suited to signals with sinusoidal components. The synthetic sine datasets were not intended to favor our method but rather to offer a controlled and interpretable setting where periodicity is known, in order to evaluate the recovery of ground truth frequency information. To mitigate potential bias, we also include evaluations on real-world irregular time series in Table 2. These results confirm that Lomb–Scargle conditioning remains helpful for imputation, even if the spectrum does not exhibit strong periodicity.
>
> **Figure 5:** We appreciate this observation and agree that the marginal distribution plots in Figure 5 do not, on their own, validate the model’s frequency reconstruction. The main purpose of these plots is to show that our imputed values align well with the observed data in the time-domain distributional sense (e.g. capturing the overall range and shape). For spectral validation, we rely on additional spectrum-based metrics (S-MAE) and Lomb–Scargle comparisons (Figure 4). We will clarify in the revision that Figure 5 primarily illustrates distributional consistency, while the Lomb–Scargle-based evaluations confirm how well our model preserves important frequency characteristics.
>
> **Computational Cost:** Please refer to the response to Reviewer 3 (**ZpA3**) for a detailed analysis of computational efficiency of our method.
>
> &nbsp;
> ### Code Availability
> We sincerely apologize for the delayed submission of the source code. The upload was unfortunately postponed due to an internal review process, but it is now publicly available in the anonymous repository linked in the abstract.
>
> &nbsp;
> ### Weaknesses
> We would like to highlight that Table 2 of our paper reports experiments on two real-world datasets (PhysioNet and PM2.5), demonstrating that our approach is not limited to synthetic scenarios. In these real-data settings, our method achieves consistent improvements over baseline imputation models, underscoring its practical applicability beyond the controlled environment of Table 1.
>
> Furthermore, like CSDI, our framework remains a probabilistic time series imputation method. Although we incorporate Lomb–Scargle conditioning, the underlying diffusion-based approach is unchanged. It still produces a distribution over missing values rather than a single deterministic estimate.
>
> &nbsp;
> ### Question for Authors (Motivation)
> Even though missingness can reflect the inherent sampling schedule, imputation remains a standard practice in the medical time series literature for two main reasons:
>
> - Many predictive tasks, such as patient mortality prediction, rely on uniform or complete data inputs. Empirical results across multiple papers show that having an imputed time grid often improves classification or regression performance on these tasks, with respect to grids with missing data.
>
> - A large number of existing methods and neural architectures assume regularly spaced inputs. Imputation is thus commonly used to align varied medical measurements (e.g. vitals, labs) to a single grid, making it easier to integrate or compare multiple signals and to apply standard machine learning pipelines that are not natively designed for irregular sampling.
>
> We will integrate this content into the manuscript as motivation for the task of irregular time series imputation.

---

> > ### Comment · Reviewer_y6X7 · 2025-04-04
> >
> > ## Experimental Design or Analyses
> > **Use of Sine Datasets**: I acknowledge that sine datasets are designed to provide an intuitive understanding on model’s performance. However, since the paper is titled “Irregular Time Series Imputation”, it should focus on the analysis on real-world irregular time series datasets (i.e., Table 2, 3). Table 1 is not strongly correlated to irregular time series imputation task, while taking too much spaces.
> > ## Motivation
> > **Reason 1**: The authors said “rely on uniform or complete data inputs”, “Empirical results across multiple papers” and “often improves classification or regression performance”, but did not provide any supporting materials to prove these statements. In my option, these are weak statements for the following reasons:
> > 1. Widely researched medical time series datasets are irregular time series datasets: (1) PhysioNet’2012 [1]; (2) PhysioNet’2019 [2]; (3) MIMIC III [3]; (4) MIMIC IV [4]. Some related works on these medical irregular time series datasets: [5-7]
> > 2. Error accumulation can make the prediction worse, where errors in imputed values affect subsequent predictions [8]. Therefore, imputation is not a necessary prerequisite for accurate prediction.
> >
> > [1] Silva, Ikaro, et al. “Predicting In-Hospital Mortality of ICU Patients: The PhysioNet/Computing in Cardiology Challenge 2012.” Computing in Cardiology, vol. 39, 2012, pp. 245–48.
> >
> > [2] Reyna, Matthew A., et al. “Early Prediction of Sepsis From Clinical Data: The PhysioNet/Computing in Cardiology Challenge 2019.” Critical Care Medicine, vol. 48, no. 2, Feb. 2020, p. 210.
> >
> > [3] Johnson, Alistair E. W., et al. “MIMIC-III, a Freely Accessible Critical Care Database.” Scientific Data, vol. 3, no. 1, 1, May 2016, p. 160035.
> >
> > [4] Johnson, Alistair E. W., et al. “MIMIC-IV, a Freely Accessible Electronic Health Record Dataset.” Scientific Data, vol. 10, no. 1, 1, Jan. 2023, p. 1.
> >
> > [5] Luo, Yicheng, et al. Knowledge-Empowered Dynamic Graph Network for Irregularly Sampled Medical Time Series. NeurIPS 2024.
> >
> > [6] Wu, Zhenbang, et al. An Iterative Self-Learning Framework for Medical Domain Generalization. NeurIPS 2023.
> >
> > [7] Jarrett, Daniel, et al. Clairvoyance: A Pipeline Toolkit for Medical Time Series. ICLR 2021
> >
> > [8] Wu, Sifan, et al. Adversarial Sparse Transformer for Time Series Forecasting. NeurIPS 2020
> >
> > ---
> >
> > In my opinion, this paper should be titled as “Probabilistic Time Series Imputation” or “ Incomplete Time Series Imputation” instead of “Irregular Time Series Imputation”, since the improvement (Lomb-Scargle) has no strong correlation with the unique properties of irregular time series (i.e., irregular time interval within each variable, and unaligned observations across different variables). The paper should be revised to be aligned with the task in title.

---

> > > ### Author Response · Authors · 2025-04-09
> > >
> > > Thank you for the thoughtful evaluation of our work. We would like to clarify that Lomb-Scargle does indeed support irregularly sampled time series natively. This motivated our initial decision to use the term "irregular" in the title. However, we recognize that our primary experimental focus has been on scenarios involving partially observed time series. Accordingly, we have removed "irregular" from the title and clarified across the text that our approach primarily targets missing values. We hope this revision better reflects the scope of our work and the role of Lomb-Scargle in preserving spectral information on data with missing values.
> > >
> > > **SINES DATASET**
> > >
> > > We previously mentioned the advantage of the sine dataset for providing ground truth information on the signal's spectrum. An additional advantage of this dataset that we would like to emphasize, is that it allows us to systematically explore the performance of the imputation methods across different types of missingness (POINT, SEQ, BLOCK).
> > >
> > > **MOTIVATION**
> > >
> > > We apologize for not providing specific references in our previous reply. Below, we offer clarifications and cite recent works that motivate why imputation can still be beneficial in downstream tasks.
> > >
> > > Time series imputation allows for a broader variety of models to be applied to prediction tasks on clinical data. We agree that not every scenario strictly requires imputation, some state-of-the-art models can process irregular data directly. However, imputation remains a practical necessity in many real-world pipelines because it allows to leverage methods that rely on uniformly spaced or complete inputs, rather than restricting them to specialized architectures [1]. Moreover, there is empirical evidence that high-quality imputation can boost downstream performance in tasks like mortality prediction [2,3,4,5,6]. However, imputation quality does not always correlate with gains in downstream tasks, which has motivated further research on building imputation methods that not only reconstruct missing values but also improve final predictive metrics [7,8]. These observations illustrate that while imputation may not be always required, it remains an important and active research topic, in particular to extend the pool of applicable models in clinical time series analysis.
> > >
> > > [1] Shadbahr, T., et al. "The impact of imputation quality on machine learning classifiers for datasets with missing values". Communications Medicine 3, 139 (2023).
> > >
> > > [2] Want, J., et al. "Deep Learning for Multivariate Time Series Imputation: A Survey". Arxiv, 2025.
> > >
> > > [3] Du et al. "SAITS: Self-Attention-based Imputation for Time Series". Expert Systems with Applications, 2023.
> > >
> > > [4] Du, W., et al. "TSI-Bench: Benchmarking Time Series Imputation". Arxiv, 2024.
> > >
> > > [5] J. Yoon, et al. "Estimating Missing Data in Temporal Data Streams Using Multi-Directional Recurrent Neural Networks," in IEEE Transactions on Biomedical Engineering, vol. 66, no. 5, pp. 1477-1490,2019.
> > >
> > > [6] Luo, Y., et al. "Multivariate Time Series Imputation with Generative Adversarial Networks". Neurips, 2018.
> > >
> > > [7] Wang, Z., et al. "Task-oriented Time Series Imputation Evaluation via Generalized Representers". Neurips 2024.
> > >
> > > [8] Jarrett, Daniel, et al. Clairvoyance: A Pipeline Toolkit for Medical Time Series. ICLR 2021

---

### Official Review · Reviewer_ZpA3 · 2025-03-07

**Overall Recommendation:** 5

**Summary:**

The paper proposes a novel method designed for performing time series imputation when the input data either has missing data of is not measured at equal intervals.  The use of discrete Fourier transform in this case often leads to serious artifacts in the power density spectrum.  In contrast, the power density spectrum can be more accurately estimated using the Lomb-Scargle methods.  This paper shows how this can be incorporated into a diffusion-based method to ensure the frequency information is well captured in imputed time series.  The technique is thoroughly tested and compared against strong baseline models.

**Claims And Evidence:**

The claim is that this novel approach provides better imputation of this type of time series data.  The empirical evidence appears compelling.

**Essential References Not Discussed:**

None.

**Experimental Designs Or Analyses:**

No.

**Methods And Evaluation Criteria:**

As far as I can see the evaluations are fair and rigorous.

**Other Comments Or Suggestions:**

None.

**Other Strengths And Weaknesses:**

The paper is very well written and accessible.  The contribution is novel and substantial.

**Questions For Authors:**

None.

**Relation To Broader Scientific Literature:**

The paper appears to be well versed in the literature around time series.

**Theoretical Claims:**

N/A.

---

> ### Author Rebuttal · Authors · 2025-04-01
>
> We sincerely thank the reviewer for their positive assessment and kind words about our work. We appreciate that you find the method’s contribution to be novel and substantial, and that our empirical evaluations appear fair and rigorous
>
> ---
>
> Below we include a general analysis of the computational efficiency of our method, requested by several reviewers.
>
> &nbsp;
> ### Computation Time
> We present an analysis of the computational speed of our method (LSCD). A common concern in the reviews was the increased computational cost of Lomb-Scargle with respect to FFT. Indeed, the computational complexity of FFT is $O(N\log{N})$, while Lomb-Scargle's complexity is $O(N \cdot J)$, where $N$ is the number of points in the grid, and $J$ is the number of chosen frequencies. In our method, we use $J = N$ thus the complexity of LS results $O(N^2)$. In effect, for the grid sizes used in PhysioNet and PM2.5, our differentiable implementation of **Lomb-Scargle has a computational cost that is $\mathbf{50}$ times higher than FFT** ($1.25\times10^{-2} s$ vs $2.46\times10^{-4} s$). However, the cost of this operation does not translate directly to the computation time of our LSCD model. In Tables R1 and R2 we show a comparison of training and inference computation times respectively, for the LSCD and CSDI models, evaluated on PhysioNet and PM2.5 datasets. All computations in this analysis were performed using a g5.2xlarge AWS instance (AMD EPYC 7R32 CPU, with an Nvidia A10G 24 GB GPU). As shown in the tables, **the percentage increase in computation time is approx. $9\%$ for training and $13\%$ for inference**. However, the training of our method includes a final fine-tuning stage using the spectral consistency loss $\mathcal{L}_{SCons}$ from Section 4.3, which takes $288.7\,s/ep$ for PhysioNet and $430.3\,s/ep$ for PM2.5, due to requiring running the inference pipeline as part of the computation. Considering this step together with the full training process, **our method resulted in an additional $43\%$ training time for PhysioNet and $45\%$ for PM2.5**. This analysis has been incorporated into the manuscript.
>
> **Table R1: Training time per epoch for CSDI and LSCD.**
> The last column shows the relative increase for LSCD. Measurements were averaged over $10$ epochs.
>
> |               | CSDI (s/epoch) | LSCD (s/epoch) | $\Delta$Time (%) |
> |---------------|----------------|----------------|------------------|
> | PhysioNet     | 10.30          | 11.18          | 8.5%             |
> | PM2.5         | 14.82          | 16.23          | 9.5%             |
>
> **Table R2: Inference time per batch for CSDI and LSCD (batch size = 16).**
> The last column shows the relative increase for LSCD. Measurements were averaged over $5$ batches.
>
> |               | CSDI (s/batch) | LSCD (s/batch) | $\Delta$Time (%) |
> |---------------|----------------|----------------|------------------|
> | PhysioNet     | 88.19          | 99.18          | 13.3%            |
> | PM2.5         | 69.36          | 78.58          | 12.5%            |

---

> > ### Comment · Reviewer_ZpA3 · 2025-04-02
> >
> > Thank you for your response.  I stand by my score.  For me this was a paper I enjoyed reading.  Clearly Lomb-Scargle methods have been used before (presumably by Lomb and Scargle as well as others), but I agree with the authors view that the originality comes from integrating this into a modern differentiable programming architecture.  This is a non-trivial contribution that is often underestimated.  Given the other other literature that uses this method predates deep learning, I am still of the view that integrating Lomb-Scargle into a deep learning framework is a novel contribution.

---

> > > ### Author Response · Authors · 2025-04-09
> > >
> > > We sincerely thank the reviewer for their feedback and continuous support of our work, and for acknowledging the significance of integrating Lomb-Scargle into modern differentiable architectures. We are happy to hear that the reviewer enjoyed reading our paper.

---

### Official Review · Reviewer_jzSS · 2025-03-08

**Overall Recommendation:** 2

**Summary:**

This paper introduces a novel diffusion-based time series imputation method. Specifically designed for irregularly sampled data, the proposed method leverages the Lomb-Scargle periodogram to enhance imputation performance.

**Claims And Evidence:**

Yes.

**Essential References Not Discussed:**

No.

**Experimental Designs Or Analyses:**

Yes. The experimental design is reasonable.

**Methods And Evaluation Criteria:**

Yes.

**Other Comments Or Suggestions:**

The authors could provide a theoretical analysis of how the Lomb-Scargle periodogram influences the diffusion process.

**Other Strengths And Weaknesses:**

Strengths:
1) The idea of using the Lomb-Scargle periodogram to impute unevenly distributed time series data is reasonable.

2) The proposed method is assessed on different time series datasets.

Weakness:
1) The novelty of the proposed method is limited. Although incorporating the Lomb–Scargle periodogram into the diffusion model is relatively new, the idea of using it to deal with time series data has been studied by many previous works and the proposed framework is still mainly based on CSDI with only small changes.

2) From results presented in Table 2, the proposed method did not show significant empirical performance improvement compared to other diffusion-based method that did not use spectral method, i.e., CSDI.

3)  The theoretical results regarding the effect of the Lomb-Scargle periodogram on the diffusion process are missing.

**Questions For Authors:**

How is the computational efficiency of the Lomb-Scargle periodogram compared with FFT?

**Relation To Broader Scientific Literature:**

The idea of using Lomb-Scargle Periodogram for time series modelling is not novel. See

[1] Glynn, Earl F., Jie Chen, and Arcady R. Mushegian. "Detecting periodic patterns in unevenly spaced gene expression time series using Lomb–Scargle periodograms." Bioinformatics 22, no. 3 (2006): 310-316.

[2] Ruf, T. "The Lomb-Scargle periodogram in biological rhythm research: analysis of incomplete and unequally spaced time-series." Biological Rhythm Research 30, no. 2 (1999): 178-201.

[3] Van Dongen, H. P. A., E. Olofsen, J. H. Van Hartevelt, and E. W. Kruyt. "A procedure of multiple period searching in unequally spaced time-series with the Lomb–Scargle method." Biological Rhythm Research 30, no. 2 (1999): 149-177.

**Theoretical Claims:**

No theoretical results are provided in the paper.

---

> ### Author Rebuttal · Authors · 2025-04-01
>
> We thank the reviewer for their thorough assessment of our work and for their useful comments. Below are our detailed responses, we hope they address any remaining concerns.
>
> &nbsp;
> ### Relation To Broader Scientific Literature (Novelty)
> We appreciate the presented references and recognize that the Lomb–Scargle periodogram has a longstanding history in signal processing and time series analysis. Our goal is not to claim it as a new method for time series, but rather to show how integrating a differentiable Lomb–Scargle operator into a diffusion-based architecture opens up new possibilities for modern machine learning pipelines. Unlike classical uses of Lomb–Scargle (e.g. for detecting periodicities in irregular or biological signals), our approach is the first to incorporate it into a trainable model. This enables gradient-based optimization of spectral features, allowing the network to learn periodic structure from irregular data in a way that has not been previously explored.
>
> We have also released our differentiable Lomb–Scargle implementation to encourage broader adoption of frequency-based conditioning in irregular time-series tasks. We will revise the manuscript to cite these prior works and to clarify the distinct contributions of our approach.
>
> &nbsp;
> ### Weaknesses (Table 2 Results)
> The results in Table 2 indicate that our method outperforms all other baselines, including CSDI, across all real datasets and evaluation metrics. However, CSDI consistently ranks second, with varying margins with respect to LSCD. A follow-up analysis using a one-sided Wilcoxon signed-rank test suggests that the improvement over CSDI is statistically significant with only moderate confidence ($p=0.065<0.1$).
>
> When combining both synthetic and real datasets from Tables 1 and 2, our method significantly outperforms CSDI according to a one-sided Wilcoxon signed-rank test ($p < 0.05$ for all metrics), confirming consistent improvements in imputation quality. In the following table we display the average MAE rankings of the methods for Table 1, Table 2 and the aggregate of both tables. Notably, multiple methods outperform CSDI in the synthetic datasets, while our approach remains consistently ranked first. We will incorporate this analysis as part of the supplementary material.
>
> **Table R3: Average MAE ranking per table**
>
> |        | MEAN | LERP | BRITS | GPVAE | US-GAN | TimesNet | CSDI | SAITS | ModernTCN | Ours |
> |--------|:------:|:------:|:--------:|:--------:|:---------:|:---------:|:-----:|:-------:|:-------:|:-------:|
> | Table1 | 5.9  | 9.6  | 2.6    | 6.9    | 6.8     | 8.7       | 6.1  | 2.2    | 4.6         | 1.8   |
> | Table2 | 10.0 | 4.9  | 4.0    | 7.9    | 7.5     | 7.1       | 2.0  | 3.6    | 7.0         | 1.0   |
> | Table1+Table2 | 7.2 | 8.1  | 3.0    | 7.2    | 7.0     | 8.2       | 4.9  | 2.6    | 5.3         | 1.6   |
>
> &nbsp;
> ### Weaknesses (Theoretical Results)
> As a starting point, we point out that without the spectral consistency loss, our setup is similar to [1]. The authors condition on the high and dominant frequency components of the FFT of the observed time series, whereas our technique conditions on the Lomb-Scargle periodogram. In [1], a theoretical result is presented that shows that the conditional entropy of the diffusion reverse process given the frequency representation of time series is strictly less than the conditional entropy of the reverse diffusion process without the frequency information:
>
> $$
> \mathbb{H}({\bf z}|\hat{\bf X}_t, {\bf X}^{\bf C}, {\bf C}^{\bf H}, {\bf C}^{\bf D}) < \mathbb{H}({\bf z}|\hat{\bf X}_t, {\bf X}^{\bf C}),
> $$
>
> where ${\bf z} = \hat{\bf X}_{t-1}$, ${\bf X}^{\bf C}$ denotes the time-series observation condition, and ${\bf C}^{\bf H}$ and ${\bf C}^{\bf D}$ denote the high-frequency and dominant-frequency condition, respectively. The essence behind this theoretical result is that incorporating additional conditional information reduces the entropy of the reverse process. This result extends to *any* conditional information, which means that if we condition on the Lomb-Scargle representation of the time series, it should also reduce the conditional entropy of the diffusion process. In the revision of our paper, we will include this analysis as a theoretical insight of the method.
>
> [1] X. Yang et al. *Frequency-aware Generative Models for Multivariate Time Series Imputation*. NeurIPS, 2024.
>
> &nbsp;
> ### Question (Computational Efficiency)
> We have included a detailed analysis of computational efficiency in the response to Reviewer 3 (**ZpA3**).

---

> > ### Comment · Reviewer_jzSS · 2025-04-08
> >
> > Thanks for the response.
> > Incorporating a Lomb–Scargle layer into an existing diffusion model (I assum it is CSDI) is not novel enough. And the code for implenmenting Lomb–Scargle layer via Pytorch is not hard to find. I believe that the Lomb–Scargle layers can be replaced by other silmilar types of layers, such as FFT or Koopman.Therefore, my suggestion is that the authors should explore not only the emperical but more theretical aspects of the approaches.

---

> > > ### Author Response · Authors · 2025-04-09
> > >
> > > Thank you for your feedback, we address the individual comments below.
> > >
> > > > Incorporating a Lomb–Scargle layer into an existing diffusion model (I assum it is CSDI) is not novel enough. And the code for implenmenting Lomb–Scargle layer via Pytorch is not hard to find.
> > >
> > > At the time of submission, we were unable to find any publicly available PyTorch implementation of Lomb–Scargle. Our implementation provides a compact and efficient layer that supports batching, masked inputs, and False Alarm Probability (FAP) estimation, which are essential for stable training and practical use in deep learning pipelines. To the best of our knowledge, we are the first to incorporate Lomb-Scargle as a layer in a differentiable training workflow. Moreover, we believe that our approach addresses an important problem that has not been previously addressed: how to reliably incorporate spectral information into training in scenarios with incomplete or irregular time series. We believe that making this code publicly available will constitute a valuable contribution to the community.
> > >
> > > > I believe that the Lomb–Scargle layers can be replaced by other silmilar types of layers, such as FFT or Koopman.
> > >
> > > While FFT is often used in deep learning pipelines, it implicitly assumes uniform sampling. Irregularly sampled data or missing values typically require zero-padding or interpolation, which can distort spectral estimates as shown in Figure 1. In contrast, Lomb-Scargle is specifically designed to handle irregular or incomplete time series data [1]. Replacing the LS layer for FFT would not be appropriate in the case of time series imputation, or in other learning tasks where the input time series has missing values or is irregularly sampled. Similarly, Koopman methods for irregular or incomplete time series require interpolation of the data, to a regular grid or a complex continuous-time representation [2].
> > >
> > > [1] J. VanderPlas. Understanding the Lomb–Scargle Periodogram. The Astrophysical Journal (2018).
> > > [2] I. Naiman et al. Generative Modeling of Regular and Irregular Time Series Data via Koopman VAEs. ICLR (2024).

---

### Official Review · Reviewer_gtsg · 2025-03-13

**Overall Recommendation:** 3

**Summary:**

This paper introduces Lomb–Scargle Conditioned Diffusion (LSCD), an approach for irregularly sampled time series imputation. Unlike traditional frequency-domain methods that rely on the Fast Fourier Transform (FFT), which assumes uniform sampling and requires interpolation, LSCD leverages the Lomb–Scargle periodogram to handle missing data directly in the frequency domain. The method integrates a score-based diffusion model conditioned on the Lomb–Scargle spectrum, ensuring that the imputed time series aligns with its true spectral content. To enhance performance, LSCD employs a spectral encoder and a spectral consistency loss, reinforcing the coherence between imputed series and their frequency representation. Experiments on synthetic sine wave data and two real-world datasets, PhysioNet ICU patient records and PM2.5 air quality data, demonstrate that LSCD outperforms existing baselines in both time-domain accuracy (MAE, RMSE) and spectral preservation (S-MAE).

**Claims And Evidence:**

Yes

**Essential References Not Discussed:**

Key related works on irregularly sampled time series, particularly those for regression tasks, are missing.

**Experimental Designs Or Analyses:**

Yes

**Methods And Evaluation Criteria:**

Yes

**Other Comments Or Suggestions:**

1. There is a typo: ‘Ls’ in the equation in Section 4.2.

**Other Strengths And Weaknesses:**

Strengths:
1. LSCD elegantly integrates the Lomb–Scargle periodogram into a diffusion model, allowing it to directly condition time-domain imputation on spectral information without requiring interpolation or zero-filling. This fusion preserves the frequency structure of irregularly sampled data while leveraging the powerful generative capabilities of diffusion models.
2. By providing a differentiable implementation of the Lomb–Scargle periodogram, LSCD enables end-to-end learning, making it seamlessly compatible with modern deep learning frameworks and adaptable to various time series tasks with missing or irregularly sampled data.
3. LSCD consistently outperforms both FFT-based and diffusion-based imputation methods by achieving lower MAE, RMSE, and better spectral fidelity (S-MAE).


Weaknesses:
1. This paper lacks discussion on related works addressing irregularly sampled time series, particularly for regression tasks such as imputation and forecasting [1-9] (a representative but non-exhaustive list). The authors should elaborate on the differences and advantages of their approach over these prior works and compare it with some state-of-the-art algorithms among them.
2. This work lacks an efficiency evaluation of the proposed model, which is crucial for practical applicability and broader usage.

[1] Latent ODEs for Irregularly-Sampled Time Series. NeurIPS, 2019.
[2] GRU-ODE-Bayes: Continuous modeling of sporadically-observed time series. NeurIPS, 2019.
[3] Neural Flows: Efficient Alternative to Neural ODEs. NeurIPS, 2021.
[4] Multi-time attention networks for irregularly sampled time series. ICLR, 2021.
[5] Modeling Irregular Time Series with Continuous Recurrent Units. ICML, 2022.
[6] Neural Continuous-Discrete State Space Models for Irregularly-Sampled Time Series. ICML, 2023.
[7] Modeling Temporal Data as Continuous Functions with Stochastic Process Diffusion. ICML, 2023.
[8] GraFITi: Graphs for Forecasting Irregularly Sampled Time Series. AAAI, 2024.
[9] Irregular Multivariate Time Series Forecasting: A Transformable Patching Graph Neural Networks Approach. ICML, 2024.

**Questions For Authors:**

1. Can this model impute measurements at any timestamp even without mask placeholders? Given the same observed time series, does the number of mask placeholders influence the imputation results for a specific timestamp?
2.  What are the limitations of the proposed model?

**Relation To Broader Scientific Literature:**

The study highlights the potential of Lomb–Scargle-based spectral conditioning in machine learning applications involving incomplete time series.

**Theoretical Claims:**

No proofs in the paper

---

> ### Author Rebuttal · Authors · 2025-04-01
>
> We thank the reviewer for their thorough and positive assessment of the manuscript. We appreciate the recognition of the relevance of the work, as well as the modeling and evaluation choices. Below are our detailed responses, we hope they address any remaining concerns.
>
> &nbsp;
> ### Weakness 1 (Related Work)
> We appreciate these valuable references in continuous-time modeling and specialized architectures for irregularly sampled data. In the revision of our paper, we will include an additional discussion on the comparison of score-based diffusion-based approaches and some of the aforementioned continuous-based approaches specifically w.r.t. the task of probabilistic time-series imputation. With respect to latent ODE/SDE models, a variational framework is typically used to train the model, and conditioning on a set of observed time-points to obtain the approximate posterior enables imputation by a forward pass of the learned model (filtering). More advanced variations, such as [6], perform an additional backward pass to obtain more accurate imputations (smoothing) based on all observed information. On the other hand, methods like CSDI and our method pre-define a fixed-window size for the generated time series and explicitly learn the conditional distribution $p(\mathbf{X}^{\text{mis}} | \mathbf{X}^{\text{obs}}, \mathbf{C}^{\text{obs}})$, where $\mathbf{X}^{\text{mis}}$ denotes the missing values we are trying to impute, $\mathbf{X}^{\text{obs}}$ denotes the observed samples at the time of imputation, and $\mathbf{C}^{\text{obs}}$ denotes a conditioning variable that provides additional information (based on $\mathbf{X}^{\text{obs}}$) to enhance the imputation of $\mathbf{X}^{\text{mis}}$. Examples in training are created using a mask. At inference time, filtering and smoothing are not used to impute, but a simple forward pass through the diffusion model with appropriate conditioning based on the observed values.
>
> In addition to a deeper discussion of these models, we plan to include new experimental comparisons with a subset of these continuous-time approaches (e.g. Latent ODEs [1] or DSPD-GP [7]) in the final version of the paper as additional results. This will help illustrate how the LS-based diffusion framework fares relative to advanced continuous-time baselines, highlighting both advantages (e.g. explicit spectral preservation) and trade-offs (e.g. requiring a pre-defined grid).
>
> &nbsp;
> ### Weakness 2 (Efficiency Analysis)
> Please refer to the response to Reviewer 3 (**ZpA3**) for a detailed analysis of computational efficiency of our method.
>
> &nbsp;
> ### Question 1A (Arbitrary timestamp imputation)
> LSCD, like most score-based diffusion models such as CSDI, operates on a fixed time grid and requires explicit placeholders defined by a mask to know where to perform imputation. It does not generate a continuous function that can be queried at any arbitrary timestamp. In contrast, models like Neural ODEs learn continuous latent trajectories that can be evaluated at any point in time, even those not seen during training or inference.
>
> &nbsp;
> ### Question 1B (Effect of number of mask placeholders)
> Our model, like CSDI, performs joint conditional imputation over masked timestamps, and thus the prediction at a given timestamp may depend on the number and location of other masked points. However, we additionally condition on the spectrum of the observed time series, which provides global frequency information that can help stabilize the imputation process.
>
> &nbsp;
> ### Question 2 (Limitations)
> Our approach presents the following limitations:
>
> - **Irregularly Sampled Data:** Our method supports missing time-steps but relies on a regular time grid. It also supports irregular sampling, but this requires upsampling to a fine-grained grid, which may be computationally expensive.
>
> - **No Continuous-Time Output:** Unlike neural ODEs, LSCD does not output a continuous function over time. This may limit its applicability in tasks requiring continuous-time interpolation or forecasting.
>
> - **Frequency Grid Dependence:** The model conditions on a fixed set of frequencies. An improper grid choice could lead to suboptimal conditioning, although this is mitigated by the spectral encoder and FAP-based filtering.
>
> We will incorporate the discussion of the limitations of our approach in the manuscript.

---

> > ### Comment · Reviewer_gtsg · 2025-04-04
> >
> > Based on the response regarding Q1 and Q2, this work fails to handle the imputation task for general irregularly sampled time series (i.e., interpolation in continuous time). I think the use of the term 'Irregular Time series Imputation' in this paper is not rigorous as Irregular Time series is not equivalent to time series with missing data.

---

> > > ### Author Response · Authors · 2025-04-09
> > >
> > > Thank you again for your thorough analysis of our method. We would like to provide additional clarifications regarding its limitations and how it differs from continuous-time approaches. While score-based diffusion models such as CSDI and LSCD rely on a fixed time grid, they can still handle the interpolation of irregularly sampled series. As an example, the authors of CSDI show results on irregular time series interpolation in Section 6.2 of the paper, where they compare with two continuous-time baselines (Latent ODEs [1] and mTAN [2]). Furthermore, Lomb-Scargle natively supports irregularly sampled data, hence it can be integrated in both grid-based and continuous-time approaches seamlessly. However, grid-based methods such as LSCD and CSDI only support irregularly sampled time series **provided that the interpolation time points are known at training time**. In contrast, continuous-time methods only require knowledge of the interpolation time points at inference time. We have included a discussion of this point in the text, highlighting the advantage of continuous-time methods.
> > >
> > > In Table R4, we present preliminary results for irregularly sampled time series interpolation, comparing LSCD with CSDI and two continuous-time baselines (Latent ODEs [1] and mTAN [2]). We follow the experimental setup from Section 6.2 in CSDI. Results show that LSCD handles irregular sampling effectively, with moderate improvements over CSDI, the second-best performing model. However, we acknowledge that these experiments are comparatively limited in scope, and the more extensive evaluations in Tables 1 and 2 showcase results on time series data with missing values rather than fully irregular data. Accordingly, we have removed the term "irregular" from our paper’s title, as suggested by Reviewer 4 (y6X7), and carefully revised the manuscript to clarify that we primarily address missing data rather than irregularly sampled time series.
> > >
> > > **Table R4: Results on irregularly sampled time series interpolation.**
> > >
> > > | Metric       | LatentODE*|  mTAN* | CSDI     | LSCD   |
> > > |--------------|-----------|--------|----------|--------|
> > > | MAE 10%      |   0.522   | 0.389  |  0.371   | 0.281  |
> > > | RMSE 10%     |   0.799   | 0.749  |  0.798   | 0.528  |
> > > | MAE 50%      |   0.506   | 0.422  |  0.387   | 0.382  |
> > > | RMSE 50%     |   0.783   | 0.721  |  0.687   | 0.672  |
> > > | MAE 90%      |   0.578   | 0.533  |  0.543   | 0.545  |
> > > | RMSE 90%     |   0.865   | 0.836  |  0.851   | 0.850  |
> > > (*) Values obtained from Tashiro et al. (2021) [3]
> > >
> > > [1] Latent ODEs for Irregularly-Sampled Time Series. NeurIPS, 2019.
> > > [2] Multi-time attention networks for irregularly sampled time series. ICLR, 2021.
> > > [3] CSDI: Conditional Score-based Diffusion Models for Probabilistic Time Series Imputation. NeurIPS, 2021.

---

### Decision · Program_Chairs · 2025-05-01

**Decision:**

Accept (poster)

**Comment:**

The authors propose to use Lomb-Scargle periodograms as a new layer in a diffusion model for time series. The Lomb-Scargle (LS) periodogram is a technique that allows to estimate the spectrum of a signal even in the case of very irregular sampling. The model (called LSCD) proposed by the authors is essentially a variant of the popular CSDI model (Tashiro et al., 2021) that additionally leverages LS as a layer to handle missing values. On several data sets (several synthetic and two real), this addition seems give very good results (and in particular improve the results of CSDI).

Reviewers generally agreed that using LS within a deep learning framework (and, in particular, a diffusion model) was a sensible idea. However, there was significant disagreement on whether or not the paper was worthy of ICML. Two reviewers thought that the contribution was too simple, and that the novelty was not sufficient for ICML. One reviewer remained neutral (enjoying the elegance of the use of LS but acknowledging the limited novelty), and one championed the paper, arguing that simplicity was a quality and not a shortcoming.

It is indeed true that implementing LS as a differentiable layer in modern deep learning frameworks such as PyTorch is not very hard. Even though I think both sides of the argument had sensible arguments, I do believe that simplicity is a quality in that case. Moreover, I believe that there is a small chance that this paper might make LS more popular in the machine learning world. The LS periodogram (which is essentially equivalent to fitting a sinusoidal function to the signal using the mean-squared error loss) is widely used in astrophysics (e.g. VanderPlas, 2018) but not very well-known in machine learning. Its main strength is that it can deal seamlessly with irregular time series, and could be used in many contexts for dealing with time series. Because of this small chance of having a high impact on the community, I support accepting this paper.

While I recommend acceptance, I strongly encourage the authors to take into account all comments. In particular, adding more details on the theoretical/practical consequences of using sinusoidal functions (as was done in some of the discussions with reviewers) would clearly improve the paper.

# References mentioned in the submission

- VanderPlas. Understanding the lomb–scargle periodogram. The Astrophysical Journal Supplement Series, 2018.
- Tashiro et al., CSDI: Conditional score-based diffusion models for probabilistic time series imputation, NeurIPS 2021